# Momentum, volume and investor sentiment study for u.s. technology sector stocks—A hidden markov model based principal component analysis

Shaoshu Li [ID]*

Department of Economics, Cornell University, Ithaca, United States of America

* sl2555@cornell.edu

**Citation:** Li S (2025) Momentum, volume and investor sentiment study for u.s. technology sector stocks—A hidden markov model based principal component analysis. PLoS One 20(9): e0331658. https://doi.org/10.1371/journal.pone.0331658

## Abstract

In this paper, we study the impact of momentum, volume and investor sentiment on U.S. tech sector stock returns using Principal Component Analysis-Hidden Markov Model (PCA-HMM) methodology. Price and volume are two well-known aspects in general equilibrium model. Momentum effect arises from the determination of prices in the market equilibrium. By studying momentum, volume and investor sentiment, we intuitively connect theoretical finance model with modern behavior finance topic. Instead of predicting future stock returns using machine learning models and doing comparisons, we apply the PCA-HMM method to reveal the hidden force in the financial and macroeconomic time series to calibrate different regimes. Combining the traditional financial study methods with modern machine learning techniques, we show investor sentiment effect show the primary effect on tech sector stock return which outweighs volume effect and momentum effect. The volume effect also has ineligible impact on stock return. The investor sentiment effect and volume effect show most impact on tech stocks with large or medium market shares. In contrast, momentum effect has very trivial correlation with tech sector stock return, from both stock level and individual state level. We also discuss the underlying mechanisms behind above findings according to tech sectors' unique characteristics, as well as raise risk management concerns. Using such PCA-HMM method, we reveal the unique patterns in tech sector stock returns. The PCA-HMM method can especially help us to identify those edge cases under which market behaves irregularly.

## 1. Introduction

Regime switching model has been widely used to study the time series volatilities across different states in study economic and finance series. Compared to linear time series models, regime switching model can calibrate the time series behaviors in different regimes and capture complex dynamic patterns [1]. To analyze the

**Data availability statement:** All the data is publicly accessible and can be downloaded from yahoo finance website (https://finance.yahoo.com/). The Python software contains "yfinance" library to automatically download yahoo finance data. By simply import the library and write related function, we can download data easily. See: (1) https://github.com/yahoo-finance/yahoo-finance (2) https://www.geeksforgeeks.org/python/get-financial-data-from-yahoo-finance-with-python/ (3) https://www.analyticsvidhya.com/blog/2021/06/download-financial-dataset-using-yahoo-finance-in-python-a-complete-guide/ (4) https://www.geeksforgeeks.org/python/how-to-use-yfinance-api-with-python/ (5) https://www.kaggle.com/code/himanshunakrani/exploring-yfinance-yahoo-finance-python-library.

**Funding:** The author(s) received no specific funding for this work.

**Competing interests:** The authors have declared that no competing interests exist.

economic and finance data series structures using regime switching model has been an everlasting research topic. In Hamilton [2], he theoretically proposed the Markov regime switching model in analyzing regime changes and business circles. The Markov regime switching model can make reasonable estimation of the probabilistic inference of the shifts in an economic series. Author applied this model to perform empirical study the U.S. post war real GNP data. In Schaller and Norden [3], authors showed the stock market returns could be predicted by the price/dividend ratio using the Markov regime switching model. They also showed the transition probabilities of Markov regime switching model varied over time in response to the economic variables. The response of stock returns to past price/dividend ratio was asymmetric between low return and high return states. In Ang and Timmermann [4], authors discussed the estimations and statistical properties of regime switching models in the financial market. The regime switching means, volatilities, skew, kurtosis, autocorrelations, and cross-covariances of asset returns varied across different regimes. They also characterized several aspects in equilibrium asset pricing models such as equity returns, interest rates, exchange rates and the optimal asset allocations. They pointed out the shifts of regimes from one to another were caused by the changes of economic policy and changes of investors' expectations. In Balcilar et al. [5], authors used Markov switching vector error correction (MS-VEC) models to develop the relationship between U.S. crude oil and stock market prices using monthly data from 1859 to 2013. They constructed a two-regime model according to the volatilities of the oil and stock prices. They found out the high volatilities' regime dominated before Great recession and after 1973 oil price shock caused by the Organization of Petroleum Exporting Countries, and the low volatilities regime dominated between Great recession and 1973 oil price shock. They also analyzed the relationship between high volatility regime and economic recession, as well as the impulse response for stock price against the shock of the oil price. In Wang et al. [6], authors applied Markov regime switching model to analyze Chinese mainland and Hong Kong stock markets. They divided the stock market into bear and bull regimes, and measured ambiguity degrees under different stock market regimes. They applied regressions and analyzed the impact of risk volatility factors on excess returns under both higher ambiguity and lower ambiguity degrees. The empirical results well revealed the significant difference between the bear and bull market regimes.

With the aid of emerging machine learning and artificial intelligence algorithms, we can automate the learning process of the underlying dataset and improve the prediction accuracies for the regime switching model. Rather than estimating the transitions between two regimes, the machine learning models can encompass learning several regimes at the same time and study the regime shifts simultaneously. In this paper, we majorly use hidden Markov model-based principal component analysis (PCA-HMM) to identify and predict distinct market behaviors. PCA serves as the tool for dimension reduction for the input dataset while HMM acts as the method to uncover hidden regimes in the time series. As a signal model, HMM is well-known for its effectively recognized the underlying correlations between observations and latent process by learning speech, symbols and events [7–11]. PCA-HMM has been

widely adopted in the science and engineering fields in the past twenty years. Past studies proceeded from various angles and shed light on our research. In Liu and Chen [12], they used adaptive hidden Markov models (HMM) to perform video based-face recognition. In their adaptive HMM algorithm, PCA was used to reduce all the face images to low dimensional feature vectors which were used as observation vectors in the HMM training. They also computed the diagonal covariance matrix of all the feature vectors which also were used as input in the HMM training. Through the HMM training, the statistics of video sequences and temporal dynamics were learned. During the recognition, the temporal characteristics of the video sequence were analyzed and the likelihood scores were used to justify the recognition results based on a certain feature. In Yu [13], he used hidden Markov model and dynamic principal component analysis to access machine health degradation from normal to failure. The dynamic PCA was used to extract features from vibration signals. The HMM was used to quantify machine health states. A contribution analysis was performed to uncover the effective features which detected and accessed machine health degradation. In Saracoglu [14], he applied such hidden Markov model-based principal component analysis to perform heart disease diagnostic. He applied PCA method to perform the dimensional reductions and feature extractions of the heart sound signal data. He then used HMM to perform the classification. The empirical analysis applied this model on two heart diagnostic datasets. By comparing the PCA-HMM with simple HMM, ANN method and KNN classifier, author revealed the PCA-HMM method had improved accuracy of single HMM and the success rate of HMM classifier had surpassed the ANN and KNN. In Kouadri et al. [15], the hidden Markov model-based principal component analysis was applied to improve the fault detection and diagnosis availability, reliability and required safety for overall modern Wind Energy Conversion (WEC) systems. The PCA technique was used for extract the features to be used as input for the HMM classifier. Using simulated data from the WEC, authors revealed the overall PCA-HMM approach depicted higher accuracy and efficiency than PCA-SVM method, especially when data was heavily mixed with noise and measurement errors. In Vidaurre [16], author studied the functional connectivity between brain areas in neuroscience field by improving HMM-PCA method. He showed the improved HMM-PCA method could reduce estimation errors and outperformed related estimation approaches.

Applying hidden Markov model based principal component analysis method, we study the impact of momentum, volume and investor sentiment on U.S. tech sector stock returns. Above three aspects are core concepts heavily being discussed in classical or modern financial theory. In coherent with the methodology of factor models shown in the previous literature [17–22], we study the equity return in response to above three aspects. Instead of using machine learning models to predict future stock returns or predict future stock market regimes, we apply the PCA-HMM method to reveal the hidden force in the financial and macroeconomic time series to calibrate investor sentiment effect to market regime shift. Using ten years data from 2015 to 2024, we show investor sentiment effect plays a dominant effect on tech sector stock return which outweighs volume effect and momentum effect. By comparing regressions with and without indicator variables, we show volume effect can also be largely influenced by investor sentiment effect. The level of investor sentiment effect and volume effect vary greatly across stocks and individual states with large or medium market shares stocks being more affected. In contrast, momentum effect has comparatively trivial impact on tech sector stock return. We explain these empirical findings corresponding to unique properties of tech sectors and tech stock trading patterns. The PCA-HMM method can also help us to identify those individual states or corner cases under which financial market behaves irregularly.

The main contributions of this paper can be summarized into following aspects. (1) This paper adopts the Hidden Markov Model (HMM) to study the observed data as a series of outputs depending on hidden states. Compared to the traditional Markov Regime Switching Model with rigid parametric format, the HMM model shows great flexibility in its model construction and handles a variety of observation distributions. (2) Our study improves the categorization of market states. In difference with common financial research which categorizes market regimes by bearish and bullish under most occasions, we allow the stock market depict multiple regimes. (3) We select top five principal components in our model to represent market sentiment instead of selecting the first principal component to represent market sentiment as common

practice in past sentiment type literature. Our investor sentiment calibration would address a broader class of real-world scenarios and coherent with the statistical theory methodology of principal component analysis to a larger extent. (4) We also largely improve the frequency of the data point in financial sentiment research and maximumly increase the efficiencies in financial sentiment calibration. In the past financial sentiment works, the variable selections are mostly firm level financial ratios and macroeconomic indexes, in mixture with stock market indicators. The former two types of variables are of relatively low latencies due to publications of corporate financial reports and macroeconomic census. In contrast, the data in our study are of much higher frequencies and consistent in their frequency levels. (5) Our study also links the investor sentiment impact with modern risk management framework through measuring risk aversion parameter and digging into its patterns across different states.

The paper proceeds as follows. Section 2 summarizes past literature related to momentum, volume and investor sentiment. Section 3 shows the mathematical setup for PCA-HMM method. Section 4 provides the preliminary study of the data, empirical output and result discussions. Section 5 concludes the paper and illustrates potential future research directions.

## 2. Literature review

In this paper, we study the impact of momentum, volume and investor sentiment on U.S. tech sector stock returns using PCA-HMM methodology. Price and volume are two well-known aspects in general equilibrium model. Momentum effect arises from the determination of prices in the market equilibrium. Momentum strategy represents buying stocks that have high past returns (past winners) and selling stocks that have poor past returns (past losers). Investor sentiment, on the other hand, is a behavior finance topic which analyzes investors sentiment and the way it influences stock return. The influence of these three aspects on asset returns have been heavily discussed in the past literature.

### 2.1 Momentum

Past literature calibrated momentum effect in multiple ways and establish its correlations with stock return. Jegadeesh and Titman [23] showed stock returns had momentum effect. The strategies of buying past winners and selling past losers could generate significant positive returns over 3–12 month holding periods. Moskowitz and Grinblatt [24] revealed industry stock return momentum could contribute to most stock return momentum. Industry momentum investment strategies namely buying stocks from past winning industries and selling stocks from past losing industries could be much more profitable than traditional momentum strategies. Lewellen [25] studied the momentum in stock returns by taking account of industry, size, and book-to-market factors. He concluded well diversified portfolios such as size and book-to-market portfolios could exhibit as strong momentum effect as in individual stocks and industries. George and Hwang [26] documented using 52-week high price could improve the forecasting power of future returns. Nearest to 52-week high was a much better predictor of future returns than past returns of individual stocks. The 52-week high price had predictive power regardless of individual stocks had had extreme past returns. Hong et al. [27] tested the gradual-information-diffusion model and showed the profitability of momentum strategies declined sharply with firm size. Holding firm size fixed, momentum strategies worked better among those stocks with low analyst converge. Kelly et al. [28] used conditional factor pricing model to establish the relationship of stock momentum, long-term reversal and past return characteristics with future realized beta for return predictions. The time varying latent risk factors were calibrated using instrumented principal component analysis. They showed conditional risk exposure captured by betas could be used in return predictions. Goyal et al. [29] extended past studies of momentum using U.S. data to both developed and emerging countries dataset. They showed the momentum resulted from investors underreacted to information arriving gradually. They also showed momentum profit was higher in upmarket and in low volatilities states outside U.S.

## 2.2 Volume

Previous literature documented several impacts of stock volume on the stock returns and their backward relationships from different aspects. Campbell et al. [30] found the first-order daily autocorrelation of stock returns tended to decline with volume. This phenomenon held for very large stock indexes and individual stock returns. Stock price declined on a high-volume day was more likely to associate to increase stock returns than a low-volume day. Chordia et al. [31] showed trading volume had significant role in information dissemination which was a significant determinant of cross-autocorrelation patterns in stock returns. The stock returns with high trade volume led to stock returns with low trading volume. High volume stocks adjusted faster to market wide information than low volume stocks. Gervais et al. [32] investigated the phenomenon of extreme trading activities contained information about future stock prices movement. High volume stocks tended to show positive excess returns in the following month while low volume stocks tended to negative excess returns. They concluded the such high-volume return premium was affected by the increase visibility caused by stock trading activities. The interactions of volume and momentums have also been analyzed in the past literature and we show two examples. Lee and Swaminathan [33] discussed the functions of trading volume in cross-sectional returns prediction. High (low) volume stocks experienced lower (higher) future returns. They illustrated trading volume could determine the magnitude and persistence of future price momentum. The price momentum eventually reversed and the timing of the reversal was predictable based on the volume. Medhat and Schmeling [34] found reversal and momentum coexist in the one-month short-term horizon. Low turnover stocks showed strong short-term reversal effect while high turn-over stocks exhibited almost equally strong short-term momentum. Han et al. [35] showed mispricing majorly happened for high volume stocks, which revealed stock trading volume amplified mispricing. Expected returns showed positive correlations with trading volume for underpriced stocks while showed negative correlations for overpriced stocks.

## 2.3 Investor sentiment

Past literature provided various measures in investor sentiment calibration and arrive at fruitful empirical findings in its impact on stock return. In the past literature, market-based index (e.g.,: consumer confidence index and social media sentiment index) [36–40], stock trading data and corporate financial data [40–46], or a combination of above categories were used to measure investor sentiment. Schmeling [36] used consumer confidence index as proxy for investor sentiment and showed investor sentiment had negative prediction power of stock market returns across a group of countries. High investor sentiment led to low future stock returns and vice versa. He showed impact of investor sentiment was higher for those countries with less market integrity and more prone to overreaction. Hou et al. [43] examined the dual role of investor attention in explaining price momentum and earnings momentum strategies. They used trading volume and market state as measures of investor attention for cross-sectional and time-series tests. They concluded price momentum strengthens with investor attention, but earning momentum weakened with investor attention. In the long run, price momentum profits reversed but earnings momentum profits did not. Wang et al. [37] used consumer confidence index (CCI) as proxy of investor sentiment and found a negative relationship between investor sentiment and stock return globally. They additionally showed such negative relationship was more persistent for developed markets than emerging markets. Aggarwal [47] summarized and discussed market sentiment measures shown in the past literature. Different approaches to measure market sentiment had been examined and empirical results had been compared.

Among past investor sentiment literature, principal component analysis has been used in a number of works. Baker et al. [41,42] used the first principal component of several investment sentiment measures to represent a composite sentiment index. They used the composite sentiment index to analyze the magnitudes of sentiment trends for young, small, high volatility, unprofitable stocks. Tetlock [38] developed media pessimism factor using principal component analysis for General Inquirer index from Wall Street Journal columns. He used the media pessimism factor to study stock market activities. Seok et al. [48] constructed investor sentiment index using principal component analysis of firm level stock trading

data. They studied the investor sentiment on stock return uncertainty. Most of previous works adopted the first principal component to represent investor sentiment. The limitation lies in the fact that under most situations the first principal component cannot explain the core information in the sample set. The cut-off points for total percentage of explanation power are usually required to be above 70%. We would address this issue in our PCA-HMM model and make the investor sentiment calibration consistent with the statistical methodology of principal component method.

## 3. Model and methodology

We would like to explain the key model setup and variable selections in this part. The model used in this paper contains principal components analysis and hidden Markov model. We would also explain the big picture of our empirical study using flowchart.

### 3.1 Principal component analysis (PCA)

PCA is a common technique for dimension reduction which decomposes a given dataset into a number of principal components. By applying PCA technique, the dimensionality of the original dataset can be reduced while the most useful information can be retained. In Jolliffe and Cadima [49], detail steps of performing PCA method had been detailly summarized. We have briefly summarized the main steps as follows. Given p n dimensional vectors $x_1, x_2, \cdots, x_p$, or given compact form data matrix X where X is of $n \times p$ dimension, we need to find linear combinations of the column vectors in matrix X which contains maximum variance. Such linear combinations can be expressed into following matrix format: $\sum_{j=1}^{p} a_j x_j = Xa$, where $a_1, a_2, \cdots a_p$ are constants and a is vector of $a_1, a_2, \cdots a_p$. The variance of $Xa$ is given by var(Xa) = a′Sa, where S represents sample covariance. So, the optimization problem reduced to we maximize var(Xa) (or a′Sa), subject to unit norm vector constrain a′a = 1 and secure two different linear combinations are uncorrelated cov $(Xa'_{k'}, Xa_k) = a'_{k'} Sa_k = 0$. By setting up the Lagrange function and take first order derivative with respected to a, the question reduced to following format:

$$Sa - \lambda a = 0 \iff Sa = \lambda a \tag{1}$$

In above equation, a is the eigenvector, $\lambda$ is the eigenvalue and S is the covariance matrix. The eigenvalues are the variance of the linear combinations of X, i.e.,: var(Xa) = a′Sa = $\lambda$a′a = $\lambda$. The larger the eigenvalue is, the greater variance the linear combination brings about. According to the magnitude of the eigenvalue, we can determine the number of principal components to be retained in the model. We can compute the percentage of weight attribute to each principal component by dividing specific eigenvalue by the sum of eigenvalues:

$$\pi_j = \frac{\lambda_j}{\sum_{j=1}^{p} \lambda_j} = \frac{\lambda_j}{tr(S)} \tag{2}$$

Thus, $\pi_j$ represents the explanation power of principal component j. By setup the predetermined cut-off points for total percentage of explanation power we needed, we can determine the principal components being retained in the model. Normally one chooses 70% or higher as the cut-off points to reduce information lose.

In normal PCA practice, we need to preliminarily standardize the dataset and ensure the variables magnitude are relatively comparable. In Yu et al. [50], two popular standardization methods were mentioned, which were norm standardization and mean standardization. For norm standardization, we subtract the data by the column vector mean $\mu$ and divide the column vector standard deviation $\sigma$:

$$X^* = \frac{X - \mu}{\sigma} \tag{3}$$

For mean standardization, we divide the data by the column vector mean $\mu$:

$$X^* = \frac{X}{\mu}$$

$$(4)$$

## 3.2 Hidden markov model (HMM)

Hidden Markov model has been extensively used in computational biology and biostatistics fields. In hidden Markov model, the observations $X_t$ at time t depends on hidden state variable $Z_t$. Various past works have clear illustrations of the process for hidden Markov model. We refer to Degirmenci [51] to illustrate the details steps for hidden Markov model. The observation $X_t$ is a stochastic process. Hidden state variable $Z_t$ has Markov property. In first order Markov model, and the present state $Z_t$ only depends on the past state $Z_{t-1}$. The joint distribution of the states and observations for first order HMM can be written as:

$$P(Z_{1:N}, X_{1:N}) = P(Z_1)P(X_1|Z_1)\prod_{t=2}^{N}P(Z_t|Z_{t-1})P(X_t|Z_t)$$

$$= P(Z_1)\prod_{t=2}^{N}P(Z_t|Z_{t-1})\prod_{t=1}^{N}P(X_t|Z_t)$$

$$(5)$$

An HMM can be completely determined by initial state distribution $\pi$, state transition matrix A and emission probabilities B. The initial state distribution $\pi$ describes the initial probabilities of the states at t=0, $\pi = \{\pi_1, \cdots, \pi_K\}$. The initial probabilities at all states at t=0 sum up to one:

$$\pi_i = P(Z_1 = i), \quad where \ 1 \leq i \leq K \ and \ \sum_{i=1}^{K}\pi_i = 1$$

$$(6)$$

Similar to Markov chain, the state transition matrix A is a $K \times K$ matrix where the element $a_{ij}$ is the transition probability from state $Z_{t-1}$ to $Z_t$. Each row in transition matrix should sum up to one:

$$a_{ij} = P(Z_t = j|Z_{t-1} = i), \quad where \ 1 \leq i, j \leq K$$

$$(7)$$

The emission probability matrix B is a $\Omega \times K$ matrix whose element $b_{kj}$ is the probability of making observation $X_{t,k}$ given state $Z_{t,j}$. The emission probabilities at each state $Z_t = j$ should sum up to one:

$$b_{kj} = P(X_t = k|Z_t = j), \quad where \ 1 \leq j \leq K$$

$$(8)$$

Baum-Welch algorithm is a special Expectation Maximization (EM) algorithm for solving HMM. Given hidden Markov model $\theta = (A, B, \pi)$, the Baum-Welch algorithm maximizes the probability of the observation X.

$$\theta^* = \underset{\theta}{\operatorname{argmax}}P(X|\theta) = \underset{\theta}{\operatorname{argmax}}\sum_{Z}P(X, Z|\theta)$$

$$(9)$$

Baum-Welch algorithm uses forward and backward algorithms to train the parameters of a hidden Markov model. The algorithm contains calculations of $\alpha$ and $\beta$ using forward backward algorithm, which then used as input for EM algorithm. Degirmenci [51] illustrated the detail mathematical steps of Baum-Welch algorithm especially with respected to E step and M step. We briefly summarize main steps in the following part. The forward and backward algorithms for Baum-Welch are shown as follows:

**Forward procedure:**
Forward procedure finds out the probability of making observations $x_1, \cdots, x_t$ and state at time t $Z_t$ is $i$, $\alpha_i(t) = P(X_1 = x_1, \cdots, X_t = x_t, Z_t = i|\theta)$, using iterative procedures:

(1) $\alpha_i(1) = \pi_i b_i(x_1)$

(2) $\alpha_i(t+1) = b_i(x_{t+1}) \sum_{j=1}^{K} \alpha_j(t) a_{ji}$

Backward procedure:

Backward procedure finds out the probability of making observations $x_{t+1}, \cdots, x_T$ given initial state at time t $Z_t$ is $i$, $\beta_i(t) = P(X_{t+1} = x_{t+1}, \cdots, X_T = x_T, |Z_t = i, \ \theta)$, also using iterative procedures:

$$\beta_i(T) = 1 \tag{1}$$

$$\beta_i(t) = \sum_{j=1}^{K} \beta_j(t+1) a_{ij} b_j(x_{t+1}) \tag{2}$$

The temporal variables can be estimated using Bayes' theorem (E step):

$$\gamma_i(t) = P(Z_t = i | X, \theta) = \frac{P(Z_t = i, \ X | \theta)}{P(X | \theta)} = \frac{\alpha_i(t) \beta_i(t)}{\sum_{j=1}^{K} \alpha_i(t) \beta_i(t)} \tag{10}$$

$$\xi_{ij}(t) = P(Z_t = i, \ Z_{t+1} = j | X, \theta) = \frac{P(Z_t = i, \ Z_{t+1} = j, X | \theta)}{P(X | \theta)} = \frac{\alpha_i(t) a_{ij} \beta_i(t+1) b_j(x_{t+1})}{\sum_{k=1}^{K} \sum_{w=1}^{K} \alpha_k(t) a_{kw} \beta_w(t+1) b_w(x_{t+1})} \tag{11}$$

The parameters in the hidden Markov model can be estimated using iterated procedure (M step):

$$\pi_i^* = \gamma_i(1) \tag{12}$$

$$a_{ij}^* = \frac{\sum_{t=1}^{T-1} \xi_{ij}(t)}{\sum_{t=1}^{T-1} \gamma_i(t)} \tag{13}$$

$$b_i^*(v_k) = \frac{\sum_{t=1}^{T} 1_{x_t = v_k} \gamma_i(t)}{\sum_{t=1}^{T} \gamma_i(t)} \tag{14}$$

$$\theta^{new} = (A^*, B^*, \theta^*) \tag{15}$$

### 3.3 Flowchart for our study

The big picture of our empirical study is highly summarized in Fig 1 and we will detailly explain the input data in the following section.

In this paper, we use regimes (or states) to depict investors' sentiment. As shown in past literature, investors receive information from different types of sources in order to make stock trading decisions and accordingly time series regimes can reflect investors' sentiment [43,52,53]. The investor sentiment for each individual tech stocks can be determined by mixture of financial market movement, macroeconomic shift and investors' perceptions of individual tech company's profitability. The major force in financial market movement and macroeconomic shift can be identified through a combination of major U.S. market financial market index, commodity prices, treasury bond interest rates and exchange rates between domestic currency and foreign currencies through PCA step. Combine those principal components derived from PCA step with individual stock trading patterns, we can identify the hidden regimes for each individual tech stocks via HMM step. With the calibration of investor sentiment by hidden regimes using PCA-HMM methodology, we perform following empirical studies according to the output of PCA-HMM steps.

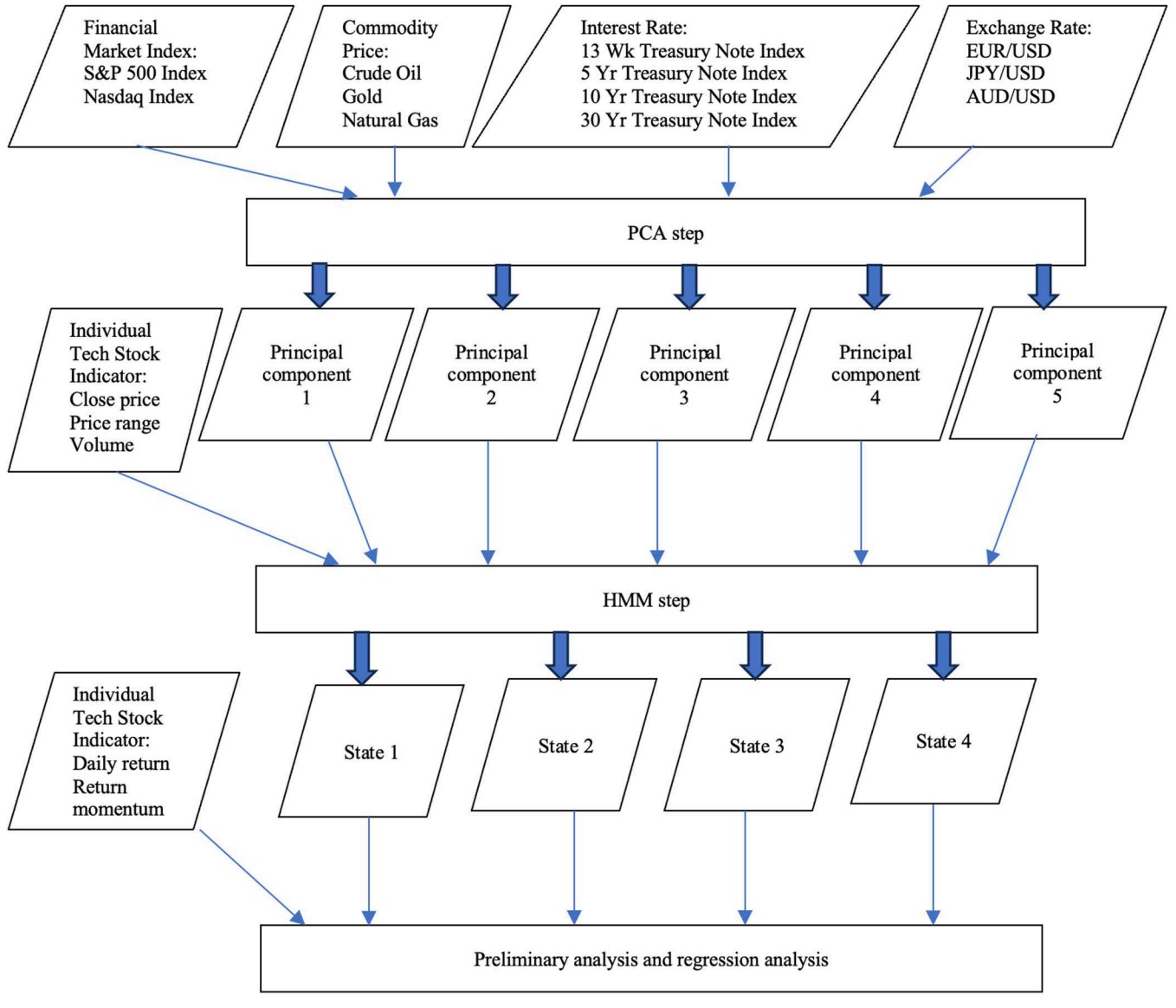

**Fig 1. Flowchart for our study.**

## 4. Empirical Analysis

### 4.1 Data

To make the data source consistent and empirical result compelling, we use ten-year data downloaded from yahoo finance ranging from January 1st, 2015 to December 31th, 2024. We use weekly data for major U.S. market stock market indexes, long-term and short-term market reference rates, commodities prices and exchange rates with standardization as the input for our PCA analysis. In the HMM analysis, we combine the principal components generated from the PCA analysis with three individual tech stocks trading indicators to construct the overall features to be used in the HMM. The three major indicators which combined with the principal components are weekly stock return, trading volume and price range. The summary of all variables involved in our study is shown in Table 1.

**Table 1. Key indicators involved.**

| General U.S. financial market conditions for technology sectors: | Frequency | Data type |
|---|---|---|
| individual tech stocks close price | weekly | stock price |
| individual tech stocks highest price | weekly | stock price |
| individual tech stocks lowest price | weekly | stock price |
| individual tech stocks volume | weekly | volume |
| Nasdaq Index | weekly | stock market index |
| S&P 500 Index | weekly | stock market index |
| 13 Wk Treasury Note Index | weekly | market reference rate |
| 5 Yr Treasury Note Index | weekly | market reference rate |
| 10 Yr Treasury Note Index | weekly | market reference rate |
| 30 Yr Treasury Note Index | weekly | market reference rate |
| Crude Oil | weekly | commodity price |
| Gold | weekly | commodity price |
| Natural Gas | weekly | commodity price |
| EUR/USD | weekly | exchange rate |
| JPY/USD | weekly | exchange rate |
| AUD/USD | weekly | exchange rate |

In our study, we haven't included companies' financial statement information nor other macroeconomics series data source. The reasons can be summarized as following two perspectives. First, most company related financial situation and future profitability have been perfected reflected in stock price. An increasing stock price related to an optimistic expectation of the investors on the financial perspectives of the company. Similarly, treasury rates, commodity prices and exchange rates are good representations of general macroeconomic data sources in stock price prediction. These three types of series are actively response to the macroeconomic regime shifts and policy changes. Second, due to most of the stock price data is of high frequency, there exists considerable frequency gap between the data source we use and company financial statement data or macroeconomic data series in Federal Reserve database. In most financial data-sets, company financial statement data is on a quarterly basis or annual basis, which cannot directly merge with weekly stock price data. Same issue happens with the macroeconomic data series. The macroeconomic data series from Federal Reserve database are mostly on quarterly, semiannual or annual basis. This also leads to a frequency mismatch between macroeconomic data series from mainstream macroeconomic database and weekly stock price data. From above two perspectives, we narrow our data source to the weekly stock market information, treasury rates, commodity prices and exchange rates from yahoo finance to maximumly expand input data availability in machine learning model and largely improve the pervasiveness of our empirical result.

### 4.2 Model development and preliminary analysis

As mentioned in previous section, the input data source for PCA step is the weekly data for major U.S. stock market indexes, long-term and short-term market reference rates, commodities prices and exchange rates. The PCA step involves construct five principal components. For most of the technology stocks in our study, these five components can explain more than 75% of the input for our PCA analysis. In the following HMM step, we combine the general stock market and macroeconomic information extracted from the PCA step with the individual tech stock trading information. To improve model accuracy, we fit the HMM using python package by assuming 1000 iterations and fixed random state number. For hidden state number selection, we generate the AIC and BIC scores corresponding to different state numbers. Due to the AIC and BIC scores are not significantly varying across different state numbers, we choose state number equals to

four where most models depict the lowest AIC and BIC scores. By performing HMM step, we classify the dataset into four states(regimes) which can perfectly take account of both individual stock performance and outside market conditions.

Use seven large tech stocks as a representation, we separately show general trends for large tech stocks close price in Fig 2, excess return, momentum and volume under individual states in Fig 3 and values for five principal components in Fig 4.

In all Figures, we use four different colors to represent four hidden states. By straightforward observation from Fig 2, most of the stock price series for large tech stocks demonstrate significant regime switching (line color switching) in early 2020. This phenomenon represents the direct COVID-19 effect. The price for large tech stocks starts to increase in the latter half of 2020, however, experiences dramatically decline in 2022. The price decline here represents the lag effects of COVID-19. The stock market recovers in 2023 and enters a high growth rate period. The rising trend for the stock price series is especially noticeable in the past two years compared to pre-COVID-19 period. Beyond the general financial market movement, the PCA-HMM methodology also identifies unique patterns for individual stocks out of the general financial market tendencies. In Fig 2, we can easily identify those few periods which have opposite colors with general color trends. In Fig 3 and 4, the return momentum, excess return, volume and five principal component values under in these periods also show great variations compared to normal time. Under the regimes behind these periods, market behaves unusually compared to normal time. Thus, using the PCA-HMM method, we can categorize the entire time horizon into different hidden states(regimes) by taking account of multiple information source and the resulting states(regimes) are rather reasonable.

## 4.3 Regression Analysis

We set up following two regressions (16) and (17) to proceed our analysis for technology sector stock momentum, volume and investor sentiment. For individual tech stock, regression (16) is performed at stock level while regression (17) is separately performed at stock level and with respected to each four individual states(regimes) calibrated by PCA-HMM. In regression (16) and (17), the dependent variable is individual tech sector stock weekly percentage return. In regression (16) and (17), the common independent variables are separately S&P 500 weekly percentage return, the momentum of stock return and the volume of each individual tech stock. The momentum of stock return is captured by autocovariance with one lag for stock percentage return in the past month (4 weeks). Investor sentiment is captured by four states(regimes) detected by PCA-HMM. The S&P 500 weekly percentage return is being controlled in these regressions due to majority of the tech stocks involved in our study are S&P 500 components. Since the stocks in our analysis are tech sector stocks which have been listed for years, we exclude other factors which can also affect stock return, namely the seasonal effect and IPO effect. We rule out the IPO effect which has been heavily discussed in investor sentiment literature in our study due to all the stocks we selected have already listed on the market for years. We also rule out the seasonal effect which can also substantially affect investor sentiment in our study. Compare with several traditional sectors such as retail, manufacture and agriculture, the tech sector has less seasonal effect. In regression (16), we also incorporate indicator variables to represent the market states identified by the PCA-HMM. These states can be interpreted as investor's sentiment to stock market movement and macroeconomic volatilities. Thus, the core regression for our investors' sentiment analysis can be shown as follows:

At stock level:

$$\text{stock return}_t = \alpha + \beta_1 \text{S\&P 500 return}_t + \beta_2 \text{momentum}_t + \beta_3 \text{volume}_t + \beta_{i+3} * \text{indicator}_i + \varepsilon_t,$$
$$\text{where indicator}_i = 1 \text{ if stock price in state } i \ (i = 1, 2, 3 \text{ or } 4), \text{ otherwise indicator}_i = 0 \quad (16)$$

At stock level and individual state:

$$\text{stock return}_t = \alpha + \beta_1 \text{S\&P 500 return}_t + \beta_2 \text{momentum}_t + \beta_3 \text{volume}_t + \varepsilon_t \quad (17)$$

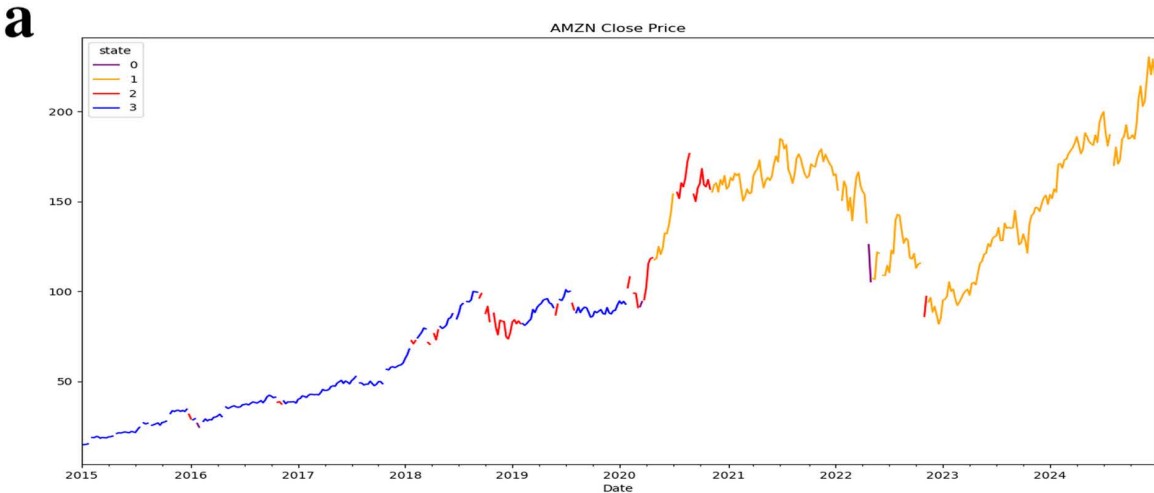

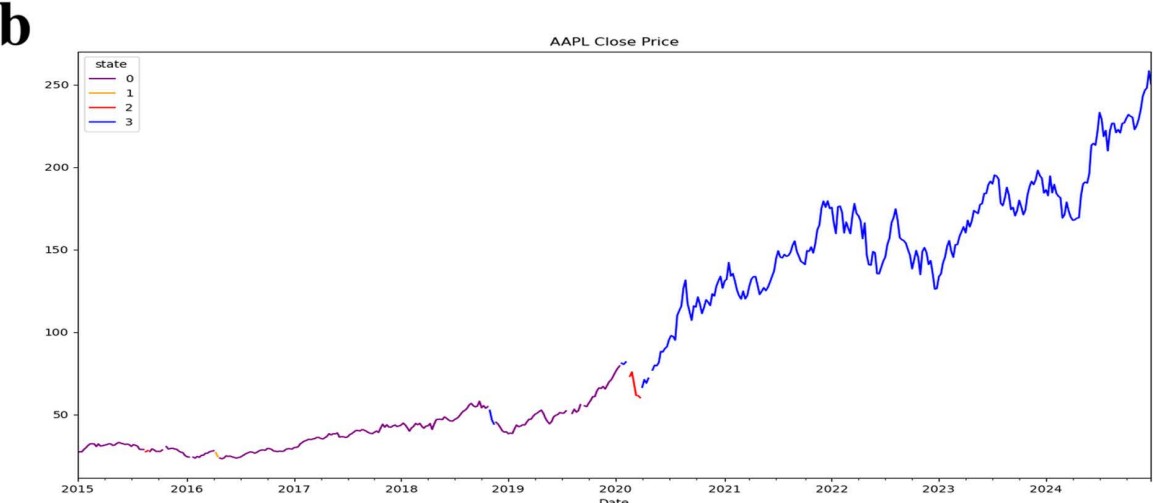

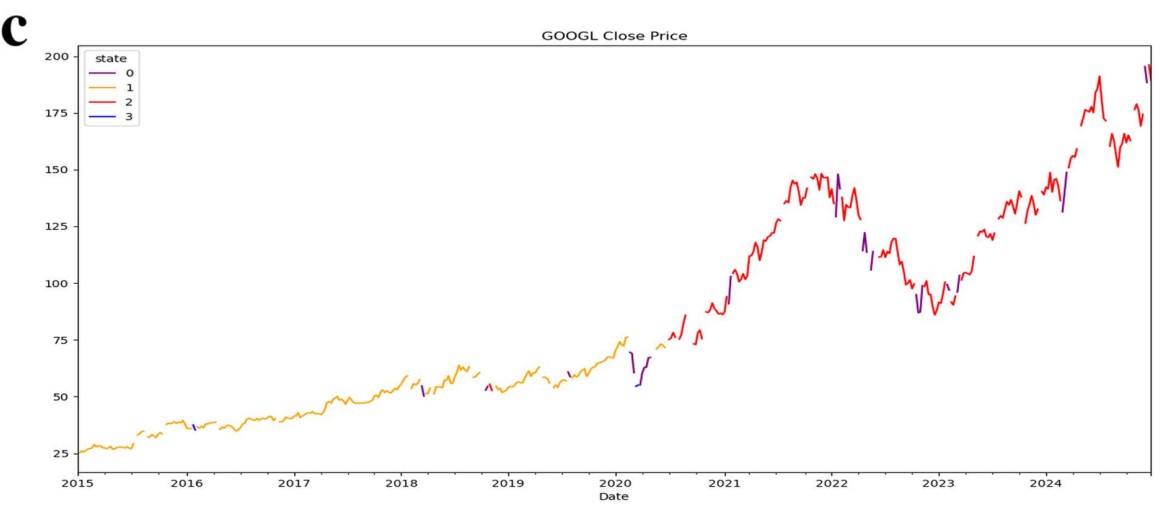

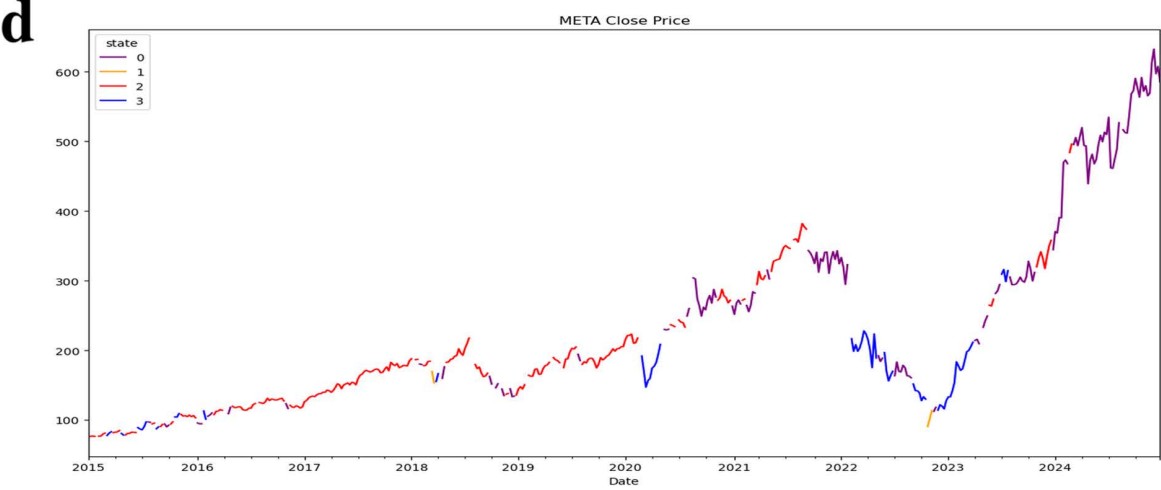

**d** METH Close Price

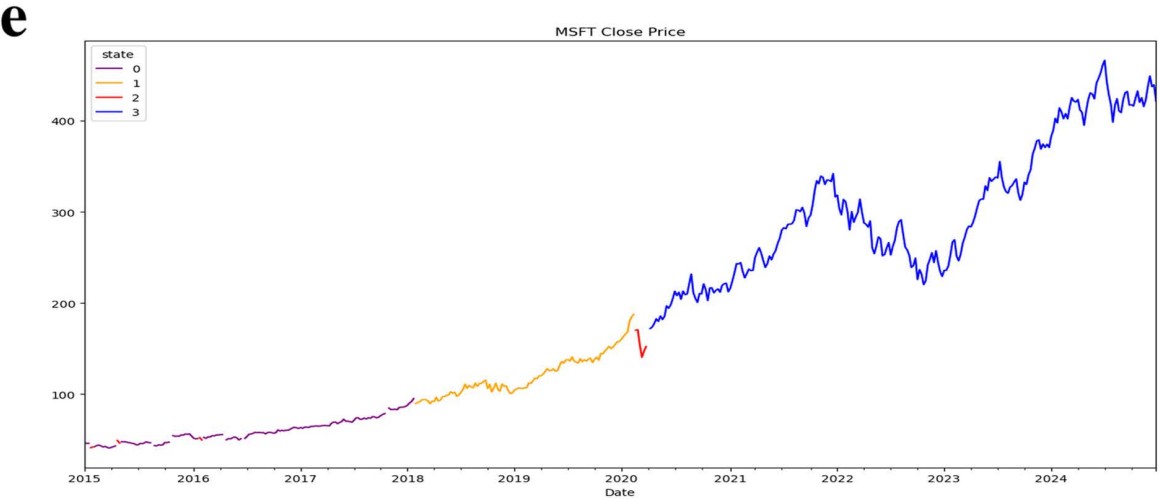

**e** MSFT Close Price

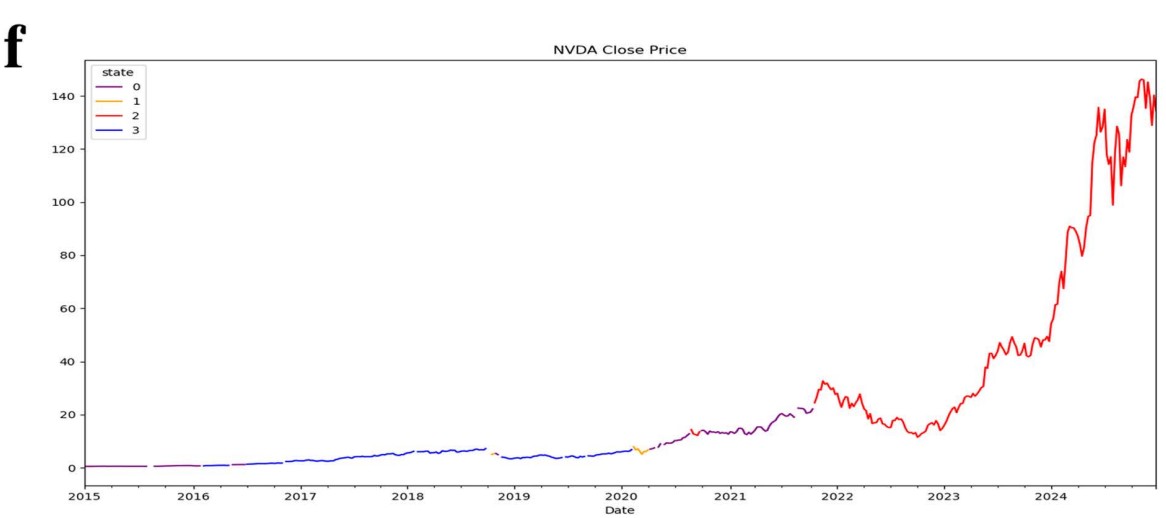

**f** NVDA Close Price

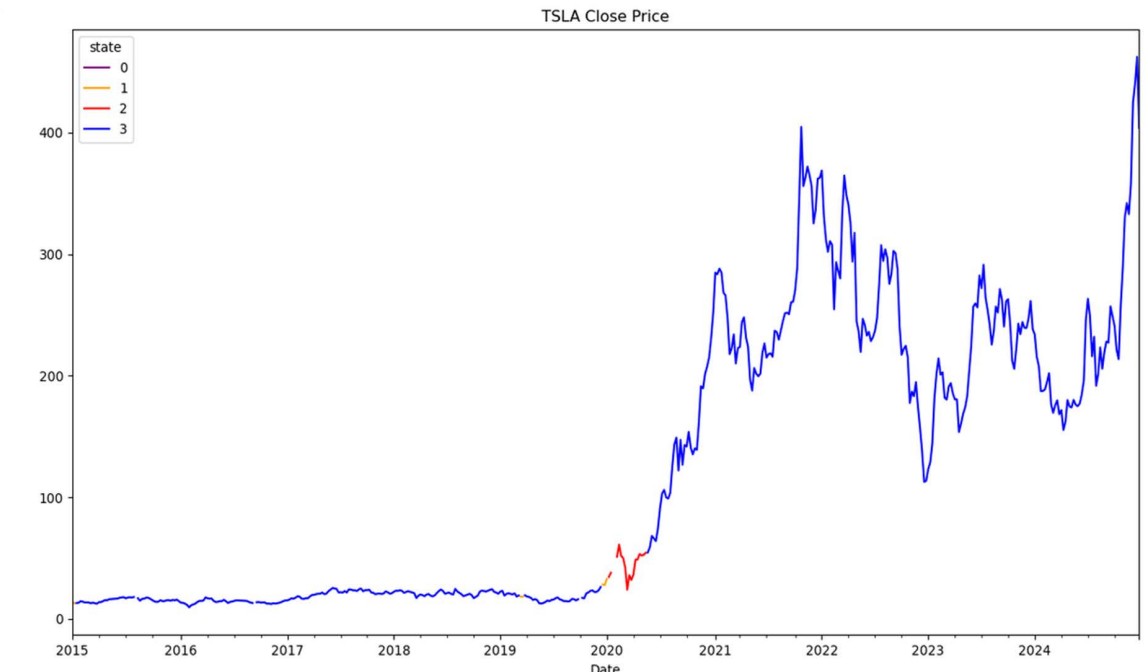

**Fig 2. General trends for large tech stock price. (a)** Stock price trend for Amazon. **(b)** Stock price trend for Apple. **(c)** Stock price trend for Google. **(d)** Stock price trend for Meta. **(d)** Stock price trend for Microsoft. **(e)** Stock price trend for Nvidia. **(g)** Stock price trend for Tesla.

We run the core regression for the seven large tech companies and companies in typical tech sectors. These sub sectors in tech industry are also categorized by yahoo finance. To expand the coverage of our study, we separately include the tech sectors which have large market shares, medium market shares and relatively small market share. We choose the companies with majority market share within each sub sector to proceed our regression analysis. For seven large tech companies, we show the descriptive statistics in Table 2, the correlations of the independent variables in Table 3 and the regression results in Table 4. For the sake of brevity, we omit display correlations and descriptive statistics for individual sub sectors. Instead, we show the regression results for individual tech sector stocks in Table 5. In both Table 4 and 5, we also list the absolute risk aversion parameter values corresponding to the overall stock level data and each individual states by assuming log normal utility function. The log normal utility shows decreasing absolute risk aversion which has commonly been adopted in solving consumption-saving problems. The empirical findings with respected to stock momentum, volume and investor sentiment can be summarized into following aspects.

**4.3.1 Momentum effect.** From Table 4 and 5, we can identify most stock returns show little stock return momentum effect for majority of the large tech companies and individual tech sectors at stock level or individual state level. In Table 4, nearly all large tech companies show no momentum effect at both stock level and individual state level. In Table 5, only a few tech stocks show certain correlations between return momentum and stock return (e.g.,: AVGO, HPQ, WDC and ARW). Compared to the volume effect and investor sentiment effect, above correlations are much more trivial (most are at 1% or 5% significance level). We also observe no strong difference for the momentum effect among sectors with large market shares and those with medium or small market shares. From the empirical output, the return momentum effect is relatively modest among different sectors and across four hidden states. As a result, the return momentum effect should be of secondary importance in determination of tech sector stock returns.

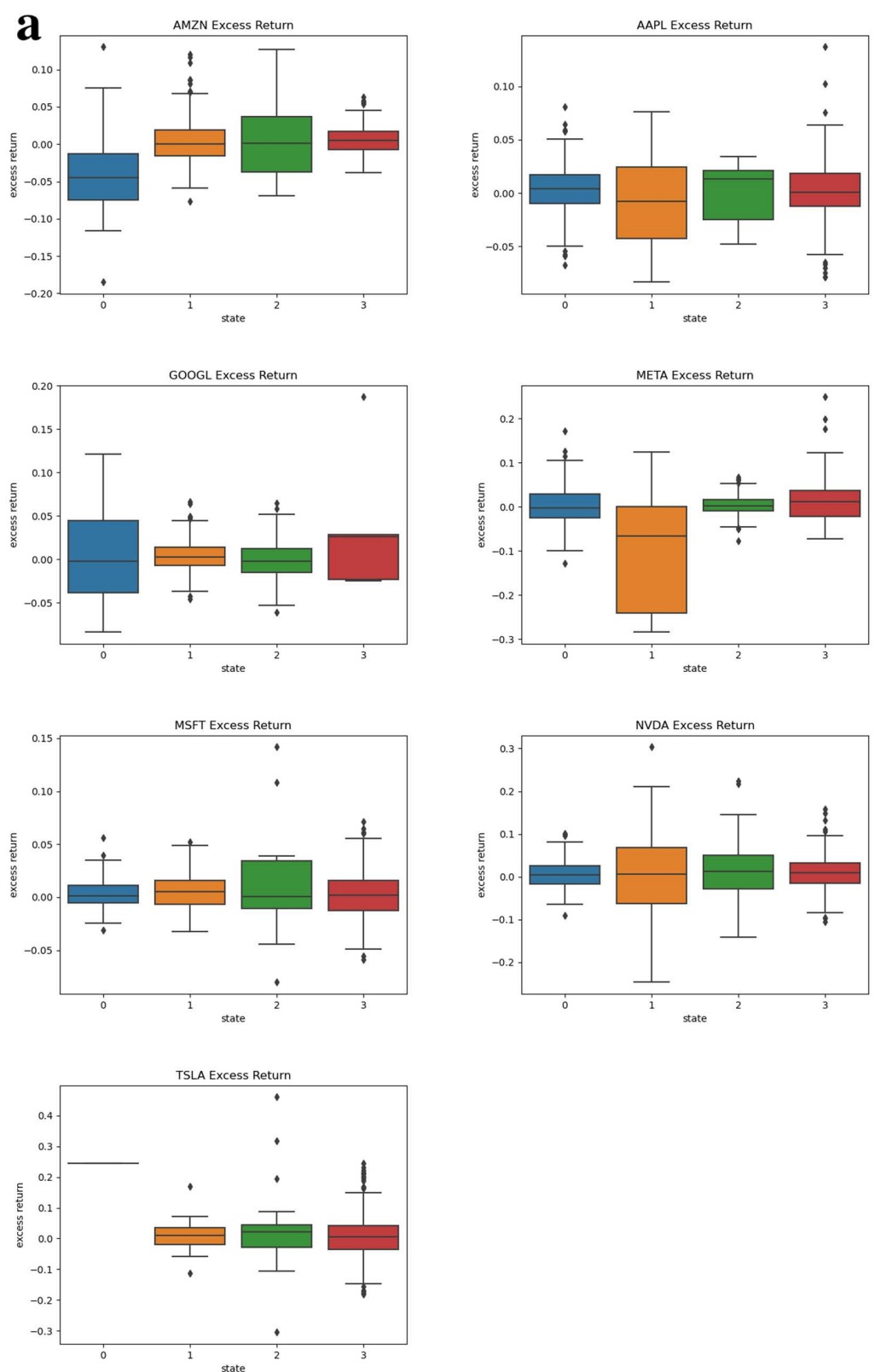

**b**

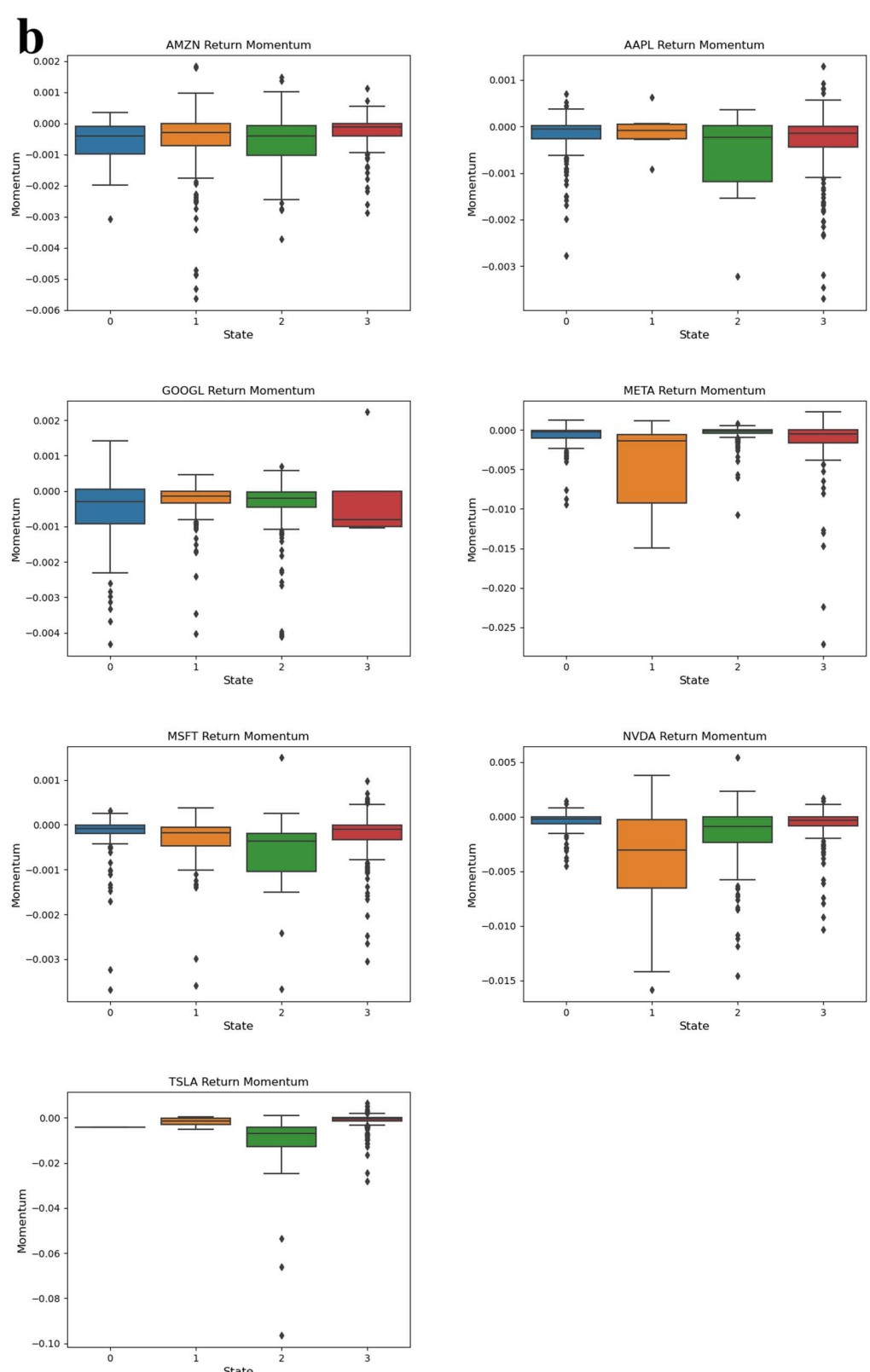

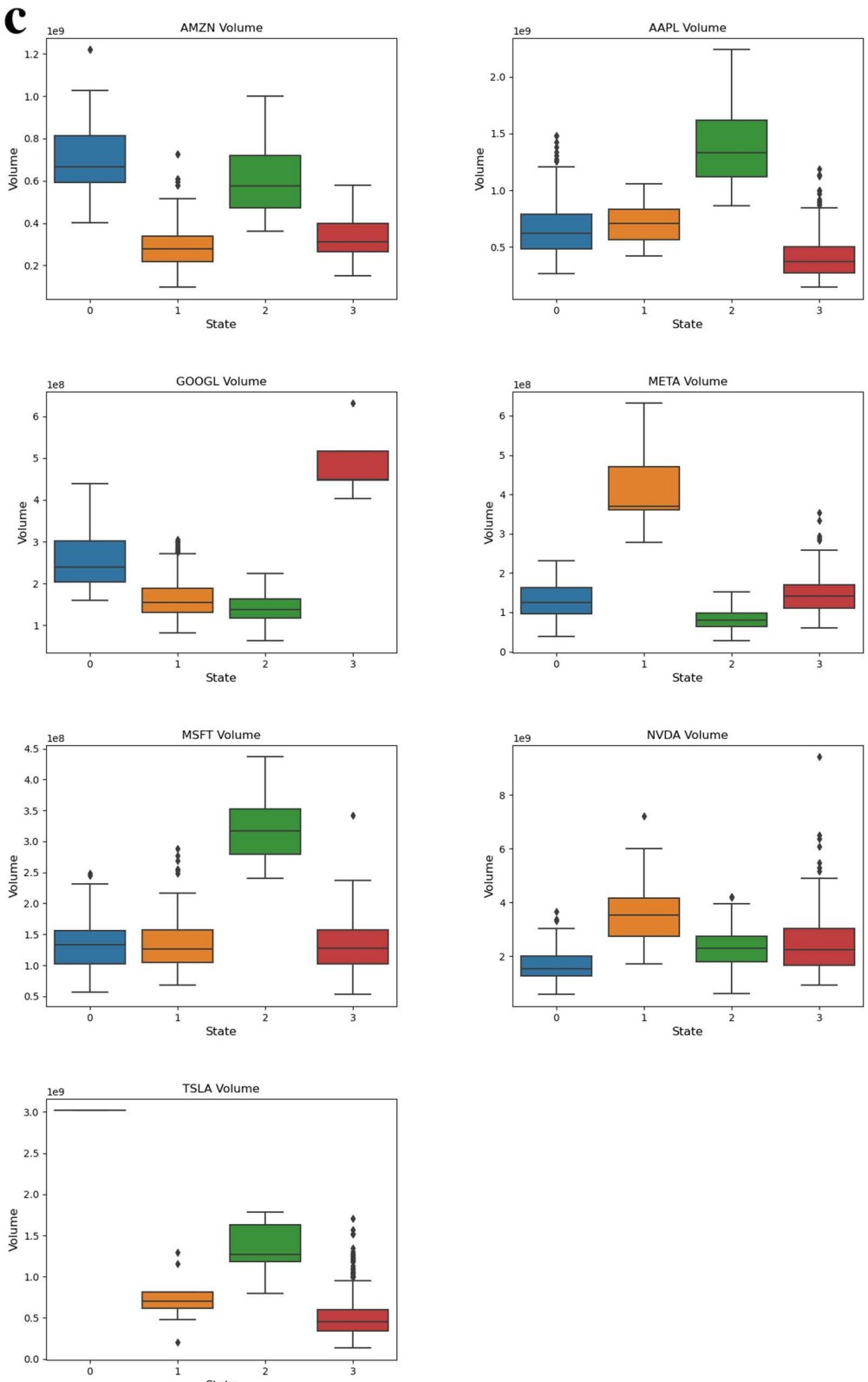

**Fig 3. Excess return, momentum and volume under individual states for large tech stocks. (a)** Box plots for large tech stocks excess return. **(b)** Box plots for large tech stocks return momentum. **(c)** Box plots for large tech stocks volume.

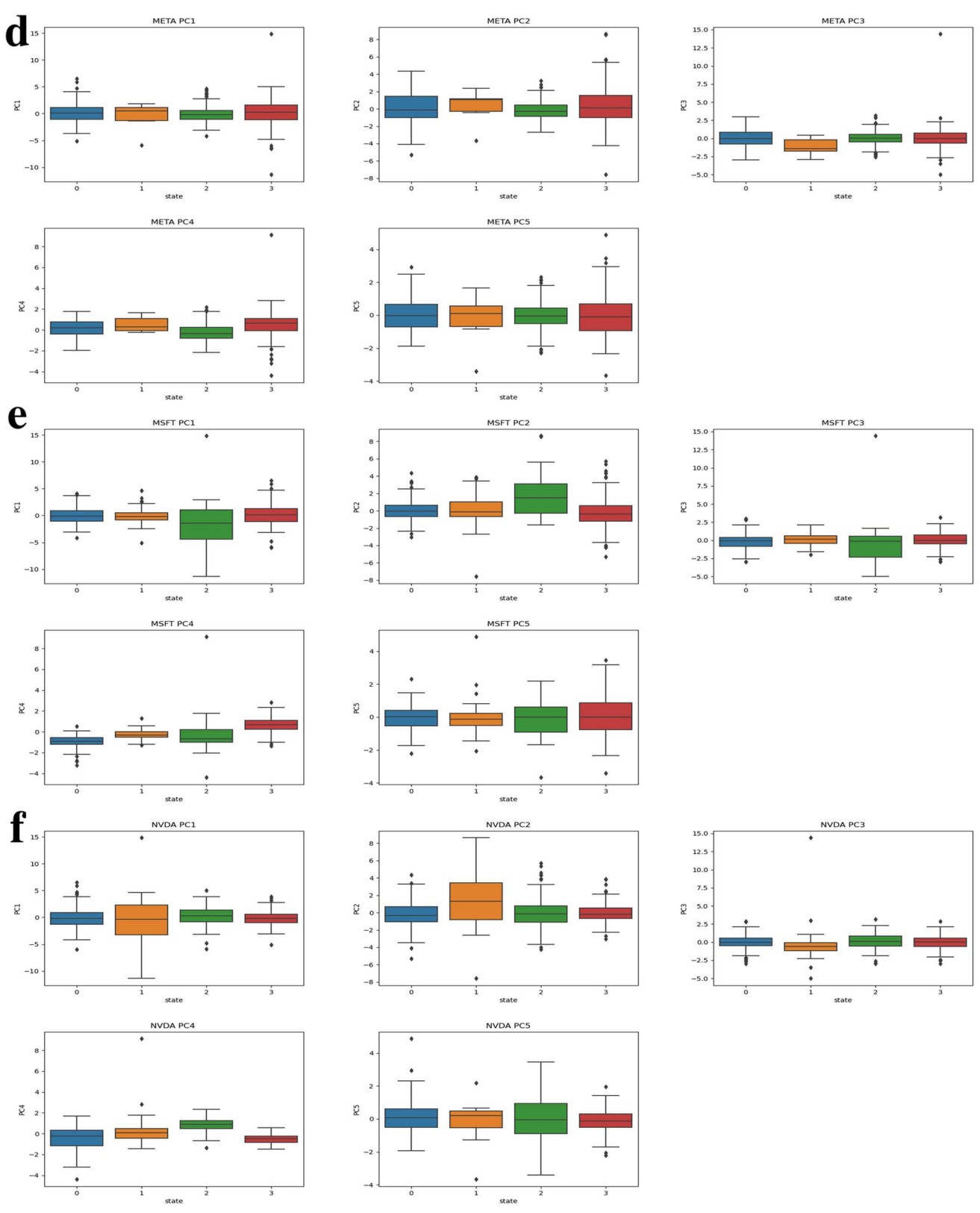

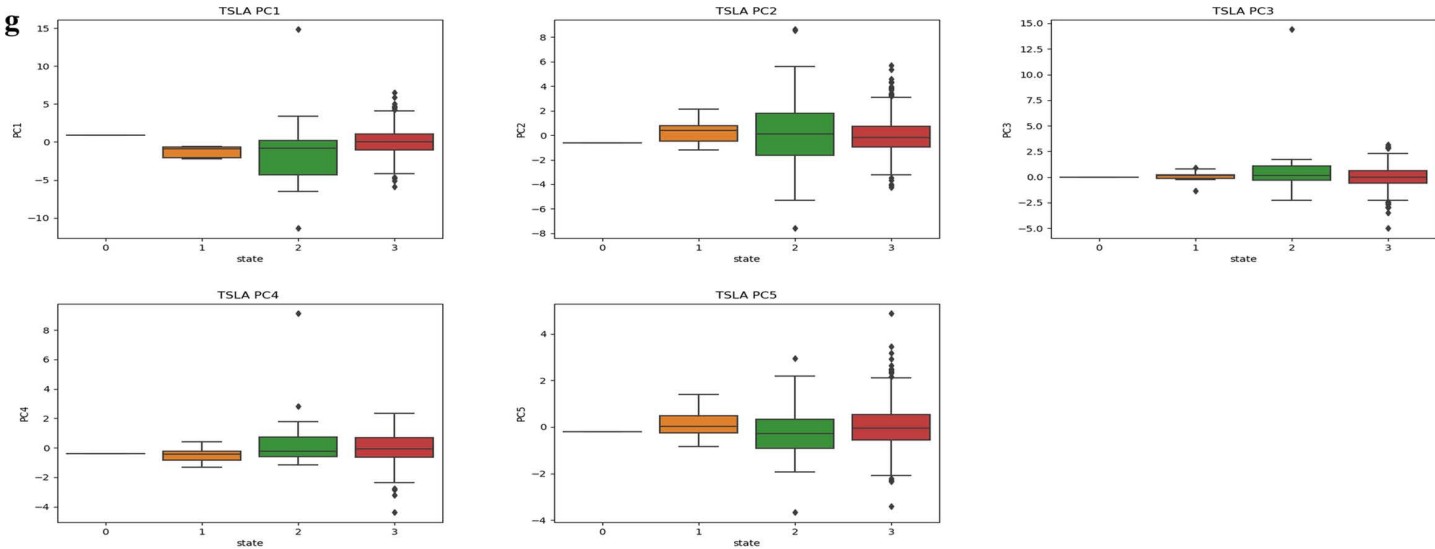

**Fig 4. Principal components under individual states for large tech stocks. (a)** Box plots for principal component values for Amazon. **(b)** Box plots for principal component values for Apple. **(c)** Box plots for principal component values for Google. **(d)** Box plots for principal component values for Meta. **(d)** Box plots for principal component values for Microsoft. **(e)** Box plots for principal component values for Nvidia. **(g)** Box plots for principal component values for Tesla.

**4.3.2 Volume effect.** As shown in Table 4, the volume effect has significant impact on most of the stock returns for large tech companies in the past years. The volume variable is significant at 1% level for the GOOGL, META, MSFT and TSLA, and significant at 5% level for NVDA at whole stock level for regression (17). Although not every individual state exposed to volume effect, volume effect exists under several individual states for all of the large tech stocks except AAPL. Apparently, the significance of the volume effect dominates the return momentum effect in determination of tech sector stock returns for most situations. Refer to both Table 4 and 5, we can also identify the volume effect is more significant for sectors with large market share than sectors with medium and small market share. For small size sector such as electronics & computer distribution, the volume variable has trivial impact on stock return at either stock level or under individual states. Overall, stock trading reflects investors' preference of active investment strategies over passively invest in stock market index. Above findings reflect the investors' keen interest of tech stocks with large market shares and sectors received more market focus. For these tech stocks, investors find incentives to actively trade these stocks which gives rise to higher volume effect. In contrast, investors behave less interest to trade small sector tech stocks which yields lower volume effect. Thus, the significance levels for the impact of volume on stock returns vary greatly across sectors and stocks with large market share sectors surpass small market share sectors.

**4.3.3 Investor sentiment effect.** Compared to regression (17), we include the indicator variables in regression (16) to express investors' sentiment to financial market volatilities, macroeconomic regime changes and individual tech stock profitability expectations. As shown in Table 4, majority of the large tech companies exposed to such investors' sentiment effect to outside market condition change under most individual states, except AAPL and TSLA. By looking into those individual states exposed to investors sentiment effect, the significance levels for the indicator variables are universally under 1%. This well reflects investors sentiment are extremely sensitive to financial market volatilities, general macroeconomic regime shifts and their own perceptions of individual stock's profitability. From Table 5, stock returns in large and medium market shares sectors are also exposed to the general market regime shift. For small share sector such as solar and electronics & computer distribution, the numbers of individual states exposed to general market regime shift

**Table 2. Summary statistics for large tech stocks close price.**

| | Mean | Std | Max | Min | N |
|---|---|---|---|---|---|
| **Panel A: Weekly return** | | | | | |
| S&P 500 | 0.00229 | 0.02215 | 0.11312 | −0.12522 | 518 |
| AMZN | 0.00574 | 0.04290 | 0.15334 | −0.20352 | 518 |
| AAPL | 0.00496 | 0.03525 | 0.15806 | −0.11757 | 518 |
| GOOGL | 0.00449 | 0.03666 | 0.19032 | −0.12355 | 518 |
| META | 0.00528 | 0.05078 | 0.27699 | −0.30257 | 518 |
| MSFT | 0.00524 | 0.03092 | 0.14364 | −0.10148 | 518 |
| NVDA | 0.01301 | 0.06499 | 0.30974 | −0.26613 | 518 |
| TSLA | 0.00975 | 0.08186 | 0.49285 | −0.43045 | 518 |
| **Panel B: Return momentum** | | | | | |
| AMZN | −0.00044 | 0.00084 | 0.00182 | −0.00563 | 518 |
| AAPL | −0.00025 | 0.00055 | 0.00128 | −0.00369 | 518 |
| GOOGL | −0.00035 | 0.00073 | 0.00222 | −0.00432 | 518 |
| META | −0.00081 | 0.00236 | 0.00226 | −0.02715 | 518 |
| MSFT | −0.00026 | 0.00053 | 0.00149 | −0.00368 | 518 |
| NVDA | −0.00110 | 0.00222 | 0.00539 | −0.01587 | 518 |
| TSLA | −0.00168 | 0.00633 | 0.00645 | −0.09652 | 518 |
| **Panel C: Trading volume** | | | | | |
| AMZN | $3.65259 \times 10^8$ | $1.62383 \times 10^8$ | $1.21931 \times 10^9$ | $9.66547 \times 10^7$ | 518 |
| AAPL | $5.57479 \times 10^8$ | $2.84576 \times 10^8$ | $2.23743 \times 10^9$ | $1.44630 \times 10^8$ | 518 |
| GOOGL | $1.68626 \times 10^8$ | $6.59291 \times 10^7$ | $6.30944 \times 10^8$ | $6.26696 \times 10^7$ | 518 |
| META | $1.09957 \times 10^8$ | $5.98631 \times 10^7$ | $6.31617 \times 10^8$ | $2.72110 \times 10^7$ | 518 |
| MSFT | $1.38320 \times 10^8$ | $5.21000 \times 10^7$ | $4.36512 \times 10^8$ | $5.27171 \times 10^7$ | 518 |
| NVDA | $2.26341 \times 10^9$ | $1.01957 \times 10^9$ | $9.40934 \times 10^9$ | $5.78476 \times 10^8$ | 518 |
| TSLA | $5.43116 \times 10^8$ | $3.10754 \times 10^8$ | $3.01728 \times 10^9$ | $1.30702 \times 10^8$ | 518 |

and the corresponding significance level for these individual states are comparability smaller than those of the large and middle size sectors. Compare investor sentiment to general market regime shift with volume effect discussed in previous section, Table 4 and 5 reflect investor sentiment to general market regime shift dominates the volume effect under most cases. Here we show two evidences separately for large tech companies and individual sectors. Compare regression (16) to regression (17), some large tech stocks (namely GOOGL, META and NVDA) show significant volume effect in regression (17). However, once state variables are included, such significant volume effect disappear. For regression result in Table 5, volume effect only comes into play once state variables are included. Above findings reveal the investors sentiment to market regime dominates the volume effect in stock return determination for both large tech companies and individual sectors.

**4.3.4 Discussions and real-world implications.** The findings in this paper are consistent and coherent with the unique properties of tech sector stock trading patterns. From our empirical findings, momentum effect is very slight, and volume effect dominates momentum effect especially for tech stocks with large and medium market shares. This well reflects tech sector stocks' high liquidity properties. Tech stocks with large market shares are highly traded by retail investors. Those retail investors who are seeking arbitrage profit via chasing price movement would buy and sell stocks on a frequently basis. This leads to large changes in trading volume. Besides, large tech stocks are heavily traded by institutional investors such as market makers and high frequency trading firms, which provides affluent liquidity source for

**Table 3. Variable correlation matrix for large tech stocks.**

| | S&P 500 return | return momentum | volume | | S&P 500 return | return momentum | volume |
|---|---|---|---|---|---|---|---|
| **AMZN** | | | | **AAPL** | | | |
| **S&P 500 return** | 1 | | | **S&P 500 return** | 1 | | |
| **return momentum** | −0.023 | 1 | | **return momentum** | 0.059 | 1 | |
| **volume** | −0.15 | −0.051 | 1 | **volume** | −0.18 | −0.082 | 1 |
| **GOOGL** | | | | **META** | | | |
| **S&P 500 return** | 1 | | | **S&P 500 return** | 1 | | |
| **return momentum** | 0.07 | 1 | | **return momentum** | −0.028 | 1 | |
| **volume** | −0.14 | −0.025 | 1 | **volume** | −0.099 | −0.14 | 1 |
| **MSFT** | | | | **NVDA** | | | |
| **S&P 500 return** | 1 | | | **S&P 500 return** | 1 | | |
| **return momentum** | 0.041 | 1 | | **return momentum** | 0.072 | 1 | |
| **volume** | −0.22 | −0.18 | 1 | **volume** | −0.061 | −0.11 | 1 |
| **TSLA** | | | | | | | |
| **S&P 500 return** | 1 | | | | | | |
| **return momentum** | −0.087 | 1 | | | | | |
| **volume** | 0.017 | −0.22 | 1 | | | | |

these stocks. Some institutional investors can provide continuous bid-ask quotes and narrow the bid-ask spread. Some institutional investors may adopt high frequency trading strategies which can also leads to large change in volume. All of these reasons explain significant volume effect for tech stocks compared to momentum effect.

Another aspect of our findings lies in that the investor sentiment effect show the most significant impact on stock returns for stock holdings, even dominates volume effect. The major difference between volume effect and investor sentiment effect lies in their time horizons. The volume effect can be contributed by the liquidity of the stock which indicates the conversions of stock positions into cash. Since tech stocks are of high liquidity, the volume effect is more of short-term horizon. On the contrary, investor sentiment effect is based on investor's perception of general macroeconomic conditions, financial market volatilities and companies own financial performance. Such perception is mostly based on psychological level which can persist in a long time period. As a result, the investor sentiment is more of long-term horizon. Our findings depict outside of detecting arbitrage opportunities in the market as volume effect depicts, investors in tech companies are more attracted by the intrinsic value of the stocks. The investors adjusted their stock holdings according to news of new product launches, government policy guidelines and quarterly financial statements. In this way, they behave more inclined to long-term horizon value investors instead of simple arbitragers. This inevitably leads to a rather strong investor sentiment effect for tech sector stocks.

We introduce the risk aversion parameter for log normal utility function in Table 4 and 5 to approximate the risk level at stock general level and four PCA-HMM states. Our empirical findings depict stronger volume effect for tech sector stocks is mostly accompanied with larger risk aversion parameter, which is held for both stock level and individual state level. Especially for states with high significance in their volume effect coefficient (at 1% significance level), the corresponding risk aversion parameters are much higher than other risk aversion parameters within the same company. Investors' risk aversion escalates and market risk arises under these market regimes. High risk aversion parameter can represent several situations such as fundamentals worse off, market panic and crisis, liquidity stress and credit crunches, etc. [54–56]. From the risk management perspective, the PCA-HMM methodology can help us identify extreme market conditions in order to control market risk.

**Table 4. Regression output for large tech stocks.**

| | S&P 500 return | return momentum | volume | constant | state 1 | state 2 | state 3 | state 4 | adjusted R squared | $\gamma$ | N for each state |
|---|---|---|---|---|---|---|---|---|---|---|---|
| AMZN | 1.1607*** | −0.2381 | $4.393 \times 10^{-11}$*** | -0.0218*** | −0.0471*** | 0.0119*** | 0.0013 | 0.0120*** | 0.431 | | 518 |
| AMZN | 1.2327*** | 0.4013 | $9.259 \times 10^{-12}$ | -0.0003 | | | | | 0.397 | 0.0153 | 518 |
| AMZN 1 | 0.5192 | −13.6144 | $2.636 \times 10^{-11}$ | -0.0846 | | | | | −0.029 | 0.0189 | 15 |
| AMZN 2 | 1.3968*** | 0.5897 | $5.25 \times 10^{-11}$** | -0.0128** | | | | | 0.462 | 0.0069 | 219 |
| AMZN 3 | 1.2180*** | −0.5439 | $5.773 \times 10^{-11}$ | -0.0289 | | | | | 0.396 | 0.0133 | 72 |
| AMZN 4 | 1.1335*** | 1.4317 | $3.92 \times 10^{-11}$*** | -0.0077 | | | | | 0.339 | 0.0242 | 212 |
| AAPL | 1.0817*** | 0.5415 | $-5.307 \times 10^{-13}$ | 0.0005 | 0.0037 | −0.0068 | 0.0021 | 0.0014 | 0.461 | | 518 |
| AAPL | 1.0856*** | 0.8070 | $9.541 \times 10^{-13}$ | 0.0021 | | | | | 0.462 | 0.0187 | 518 |
| AAPL 1 | 1.0828*** | −3.6666 | $-6.632 \times 10^{-12}$ | 0.0075* | | | | | 0.369 | 0.0299 | 247 |
| AAPL 2 | −2.8352* | −33.7425 | $-1.326 \times 10^{-10}$ | 0.0618 | | | | | 0.762 | 0.0251 | 6 |
| AAPL 3 | 0.7648*** | 4.4037 | $2.554 \times 10^{-11}$ | -0.0408 | | | | | 0.698 | 0.0253 | 11 |
| AAPL 4 | 1.1603*** | 3.2675 | $7.11 \times 10^{-12}$ | -0.0007 | | | | | 0.507 | 0.0070 | 254 |
| GOOGL | 1.1700*** | 0.8977 | $1.373 \times 10^{-12}$ | 0.0099* | −0.0041 | −0.0072** | −0.0113*** | 0.0325*** | 0.485 | | 518 |
| GOOGL | 1.1565*** | 1.0433 | $4.775 \times 10^{-11}$*** | -0.0058* | | | | | 0.479 | 0.0161 | 518 |
| GOOGL 1 | 1.3091*** | 1.0510 | $-1.354 \times 10^{-10}$ | 0.0414 | | | | | 0.524 | 0.0130 | 65 |
| GOOGL 2 | 1.0721*** | −2.4325 | $2.193 \times 10^{-11}$ | -0.0012 | | | | | 0.403 | 0.0228 | 246 |
| GOOGL 3 | 1.0753*** | 1.7919 | $-1.945 \times 10^{-11}$ | 0.0023 | | | | | 0.504 | 0.0085 | 202 |
| GOOGL 4 | 1.2608 | −39.3665 | $1.083 \times 10^{-9}$* | -0.5122* | | | | | 0.957 | 0.0241 | 5 |
| META | 1.3197*** | 0.9048 | $1.931 \times 10^{-11}$ | -0.0185 ** | 0.0193*** | −0.0864*** | 0.0186*** | 0.0299*** | 0.382 | | 518 |
| META | 1.2881*** | 0.6951 | $-8.105 \times 10^{-11}$*** | 0.0118*** | | | | | 0.333 | 0.0057 | 518 |
| META 1 | 1.2906*** | −1.0724 | $2.552 \times 10^{-11}$ | -0.0016 | | | | | 0.335 | 0.0044 | 147 |
| META 2 | 0.2600 | −5.6505 | $-1.029 \times 10^{-9}$ | 0.3126 | | | | | 0.428 | 0.0075 | 7 |
| META 3 | 1.0397*** | 2.4708** | $4.262 \times 10^{-11}$ | 0.0003 | | | | | 0.272 | 0.0061 | 273 |
| META 4 | 1.3815*** | 2.0257* | $3.854 \times 10^{-10}$*** | -0.0431** | | | | | 0.514 | 0.0066 | 91 |
| MSFT | 1.0629*** | 1.9075 | $5.847 \times 10^{-11}$*** | -0.0029 | −0.0013 | 0.0007 | 0.0011 | −0.0034** | 0.553 | | 518 |
| MSFT | 1.0609*** | 1.7722 | $6.484 \times 10^{-11}$*** | -0.0057** | | | | | 0.553 | 0.0098 | 518 |
| MSFT 1 | 1.0669*** | 1.5146 | $3.869 \times 10^{-11}$ | -0.0018 | | | | | 0.522 | 0.0197 | 147 |
| MSFT 2 | 1.1481*** | −1.6139 | $1.265 \times 10^{-11}$ | 0.0026 | | | | | 0.651 | 0.0091 | 109 |
| MSFT 3 | 1.1203*** | 5.0826 | $5.174 \times 10^{-10}$ | -0.1472 | | | | | 0.536 | 0.0176 | 14 |
| MSFT 4 | 1.0170*** | 5.2179* | $5.358 \times 10^{-11}$ | -0.0048 | | | | | 0.522 | 0.0036 | 248 |
| NVDA | 1.9372*** | 1.4888 | $3.037 \times 10^{-12}$ | 0.0079 | −0.0092** | 0.0262*** | −0.0011 | −0.0080** | 0.417 | | 518 |
| NVDA | 1.8787*** | 0.7547 | $4.799 \times 10^{-12}$** | -0.0013 | | | | | 0.409 | 0.3637 | 518 |
| NVDA 1 | 1.3471*** | −0.3807 | $8.329 \times 10^{-12}$ | -0.0088 | | | | | 0.356 | 0.7906 | 130 |
| NVDA 2 | 1.7146*** | 2.2038 | $9.291 \times 10^{-13}$ | 0.0396 | | | | | 0.283 | 0.4373 | 22 |
| NVDA 3 | 2.3110*** | 2.7638* | $1.193 \times 10^{-11}$** | -0.0119 | | | | | 0.534 | 0.0339 | 174 |
| NVDA 4 | 2.0186*** | −0.4620 | $-1.105 \times 10^{-13}$ | 0.0061 | | | | | 0.306 | 0.3522 | 192 |
| TSLA | 1.8884*** | −0.1905 | $5.068 \times 10^{-11}$*** | -0.0006 | 0.0767 | −0.0263 | −0.0294 | −0.0216 | 0.306 | | 518 |

*(Continued)*

| | S&P 500 return | return momentum | volume | constant | state 1 | state 2 | state 3 | state 4 | adjusted R squared | $\gamma$ | N for each state |
|---|---|---|---|---|---|---|---|---|---|---|---|
| TSLA | 1.9008*** | −0.0417 | $5.391 \times 10^{-11}$*** | -0.0240 *** | | | | | 0.307 | 0.03225 | 518 |
| TSLA 1 | Nan | Nan | Nan | Nan | | | | | Nan | 0.0204 | 1 |
| TSLA 2 | 5.0280 | 1.2701 | $1.891 \times 10^{-10}$** | -0.1362* | | | | | 0.457 | 0.0530 | 9 |
| TSLA 3 | 2.2585*** | −0.4139 | $1.402 \times 10^{-10}$ | -0.1536 | | | | | 0.379 | 0.0234 | 17 |
| TSLA 4 | 1.7744*** | 0.8252 | $4.473 \times 10^{-11}$*** | -0.0179*** | | | | | 0.254 | 0.0321 | 491 |

Note: *, ** and *** indicate the significance level at 10%, 5% and 1% respectively.

Our empirical findings demonstrate the PCA-HMM methodology can help market participants and regulatory party to discover those market regimes with greater volume spikes and make appropriate response. Considering the salient volume effect for large and medium market share stocks, the quant strategies for market participants should include heavy considerations of market microstructure impact in tech sector stock trading. Given high order book dynamics, the traditional momentum type strategies would be inappropriate if these strategies fail to take account of volume effect [57–59]. The market participants should closely monitor the innovation cycles, product launches and regulatory news for tech companies to monitor stock volume shifts and regularly adjusted their valuation models. On the regulatory side, the financial institutions should place greater monitors on the volume type of risk metrics for risk management purpose especially under those high-volume spike regimes.

Our empirical findings of PCA-HMM methodology can also help market participants and regulatory party to discover those market regimes with greater investor sentiment volatility and monitor market sentiment impact. The strong sentiment effect indicates asset prices can be dominantly affected by emotions for tech sector stocks beyond fundamentals. Due to high sentiment driven in tech sector, the sentiment analysis is especially meaningful for financial institutions and traders. The quant strategies should be further improved to incorporate sentiment shift of the general market and individual stocks [60–62]. Moreover, to design and incorporate diverse sentiment analytics tools and product is also of rising demand. With respect to policy and regulation, stock market becomes more volatile and unpredictable due to the sentiment swings in tech sector. Government policy should incorporate the market sentiment aspect into frameworks in order to control extreme market condition and avoid market panic.

## 5. Conclusions

Tech sector is the largest sector in U.S. economy in terms of market capitalization. Tech sector stocks are highly attractive investment choice among investors in the past years. In this paper, we study the impact of momentum, volume and investor sentiment on stock returns for U.S. tech sector stocks. Unlike traditional sectors such as retail, manufacture and agriculture, tech sector stock price is highly volatile and strongly exposed to market regime shifts. Based on the methodologies mentioned in classical literature, we make progress by combining traditional financial research methods with modern machine learning techniques. Using regime switching models to calibrate investors sentiment has been recognized in the past literature. We apply principal component analysis and hidden Markov model (PCA-HMM) methodology which has been widely used in science and engineering fields. Applying the PCA-HMM model can fully reveal the predicting power of the input data series, take account of potential hidden states in time series and let the data speak for itself. By classifying the stock return time series into individual stock return states(regimes), we can capture investor sentiment across these states by considering macroeconomic regime changes and financial market volatilities. The PCA-HMM model can especially help us to identify those edge states under which investors have unusual invest patterns compared to their regular

Table 5. Regression output for individual tech sector stocks.

| | S&P 500 return | return momentum | volume | constant | state 1 | state 2 | state 3 | state 4 | Adjusted R squared | $\gamma$ | N for each state |
|---|---|---|---|---|---|---|---|---|---|---|---|
| **Semiconductors Weight: 26.82%** | | | | | | | | | | | |
| AVGO | 1.5206*** | 1.7451** | $6.22 \times 10^{-11}$** | -0.0373*** | 0.0312*** | 0.0783*** | 0.0364*** | -0.1831*** | 0.451 | | 518 |
| AVGO | 1.4181*** | 1.3409* | $5.908 \times 10^{-11}$** | -0.0025 | | | | | 0.385 | 0.0433 | 518 |
| AVGO 1 | 1.2303*** | 1.1885 | $2.975 \times 10^{-11}$ | -0.0010 | | | | | 0.330 | 0.0573 | 328 |
| AVGO 2 | 2.4463*** | 2.7875 | $-1.518 \times 10^{-11}$ | 0.0857 | | | | | 0.494 | 0.0507 | 21 |
| AVGO 3 | 1.4026*** | 1.6042 | $6.494 \times 10^{-11}$ | -0.0011 | | | | | 0.508 | 0.0146 | 168 |
| AVGO 4 | Nan | Nan | Nan | Nan | | | | | Nan | 0.0585 | 1 |
| AMD | 1.7969*** | -0.4697 | $4.73 \times 10^{-11}$** | -0.0054 | -0.0121** | 0.0134 | -0.0107* | 0.0041 | 0.281 | | 518 |
| AMD | 1.7890*** | -0.9690 | $4.967 \times 10^{-11}$*** | -0.0093 | | | | | 0.273 | 0.0923 | 518 |
| AMD 1 | 1.9995*** | -2.6486 | $3.229 \times 10^{-11}$*** | -0.0155 | | | | | 0.498 | 0.0094 | 157 |
| AMD 2 | 1.4611*** | 3.8540 | $8.03 \times 10^{-11}$ | 0.0013 | | | | | 0.174 | 0.0225 | 33 |
| AMD 3 | 1.3721*** | -3.7196 | $1.112 \times 10^{-10}$ | -0.0301 | | | | | 0.325 | 0.0128 | 81 |
| AMD 4 | 2.0675*** | -1.0219 | $4.284 \times 10^{-11}$ | -0.0018 | | | | | 0.176 | 0.1784 | 247 |
| TXN | 1.0736*** | -0.6790 | $1.188 \times 10^{-10}$ | -0.0022 | 0.0007 | 0.0013 | -0.0031 | -0.0011 | 0.499 | | 518 |
| TXN | 1.0723*** | -0.6534 | $8.862 \times 10^{-11}$ | -0.0015 | | | | | 0.501 | 0.0114 | 518 |
| TXN 1 | 1.2381*** | 2.0973 | $3.819 \times 10^{-10}$ | -0.0082 | | | | | 0.482 | 0.0144 | 80 |
| TXN 2 | 1.0662*** | -2.7780 | $-1.745 \times 10^{-10}$ | 0.0050 | | | | | 0.332 | 0.0163 | 175 |
| TXN 3 | 0.8788** | 9.9516 | $1.174 \times 10^{-9}$ | -0.0568 | | | | | 0.274 | 0.0213 | 13 |
| TXN 4 | 1.0518*** | -1.4577 | $-3.444 \times 10^{-11}$ | 0.0002 | | | | | 0.560 | 0.0064 | 250 |
| QCOM | 1.2713*** | 0.7967 | $4.133 \times 10^{-10}$*** | -0.0236** | 0.0009 | 0.0136 | -0.0456*** | 0.0075 | 0.384 | | 518 |
| QCOM | 1.3388*** | 0.5311 | $3.067 \times 10^{-10}$*** | -0.0151*** | | | | | 0.359 | 0.0144 | 518 |
| QCOM 1 | 1.0011*** | 0.2885 | $-1.56 \times 10^{-12}$ | 0.0006 | | | | | 0.237 | 0.0211 | 220 |
| QCOM 2 | -11.8349 | -48.1913 | $5.138 \times 10^{-9}$ | -0.9542 | | | | | 0.600 | 0.0171 | 5 |
| QCOM 3 | 0.6334 | 8.8439 | $-7.214 \times 10^{-10}$ | 0.0289 | | | | | 0.261 | 0.0198 | 14 |
| QCOM 4 | 1.4669*** | 1.4717 | $4.197 \times 10^{-10}$*** | -0.0167** | | | | | 0.507 | 0.0087 | 279 |

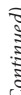

(Continued)

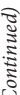

**Table 5.** (Continued)

| | S&P 500 return | return momentum | volume | constant | state 1 | state 2 | state 3 | state 4 | Adjusted R squared | $\gamma$ | N for each state |
|---|---|---|---|---|---|---|---|---|---|---|---|
| **Computer Hardware Weight: 1.95%** | | | | | | | | | | | |
| ANET | 1.2875*** | −0.7653 | $6.053\times10^{-10}$*** | −0.0796*** | 0.0549*** | −0.2084*** | 0.0139** | 0.0599*** | 0.365 | | 518 |
| ANET | 1.3311*** | −1.5864 | $-2.996\times10^{-11}$ | 0.0054 | | | | | 0.275 | 0.0869 | 518 |
| ANET 1 | 1.2249*** | 0.7835 | $2.65\times10^{-10}$** | −0.0058 | | | | | 0.251 | 0.1198 | 291 |
| ANET 2 | −31.8993 | 5904.6754 | $2.09\times10^{-8}$ | −5.3583 | | | | | -inf | 0.0660 | 4 |
| ANET 3 | 0.9778*** | −8.2741* | $1.353\times10^{-9}$*** | −0.1501*** | | | | | 0.402 | 0.1041 | 53 |
| ANET 4 | 1.5185*** | 2.3868 | $5.808\times10^{-10}$*** | −0.0156* | | | | | 0.410 | 0.0245 | 170 |
| HPQ | 1.3262*** | 2.3181** | $1.474\times10^{-10}$** | −0.0101** | 0.0077*** | | −0.0086** | 0.0058** | 0.417 | | 518 |
| HPQ | 1.3164*** | 1.9723** | $1.847\times10^{-12}$ | 0.0008 | | | | | 0.405 | 0.0595 | 518 |
| HPQ 1 | 1.1323*** | −1.2690 | $3.238\times10^{-10}$*** | −0.0112** | | | | | 0.357 | 0.0710 | 154 |
| HPQ 2 | 1.1715*** | 2.5837 | $1.307\times10^{-10}$ | −0.0224 | | | | | 0.173 | 0.0996 | 46 |
| HPQ 3 | 0.9816*** | 0.9612 | $-4.761\times10^{-10}$* | 0.0251 | | | | | 0.219 | 0.0764 | 66 |
| HPQ 4 | 1.3929*** | 2.9695** | $3.357\times10^{-10}$*** | −0.0124** | | | | | 0.514 | 0.0400 | 252 |
| NTAP | 1.0985*** | −0.2714 | $-4.378\times10^{-10}$ | 0.0116** | −0.0049 | 0.0305*** | −0.0064* | −0.0076** | 0.306 | | 514 |
| NTAP | 1.1121*** | 0.3874 | $1.324\times10^{-10}$ | −0.0003 | | | | | 0.290 | 0.0227 | 514 |
| NTAP 1 | 1.8507*** | −3.7718 | $-7.366\times10^{-11}$ | −0.0027 | | | | | 0.422 | 0.0188 | 106 |
| NTAP 2 | 1.9257* | 33.0294 | $1.353\times10^{-9}$ | 0.0034 | | | | | 0.076 | 0.0230 | 20 |
| NTAP 3 | 1.3934*** | −5.7602 | $-8.955\times10^{-10}$* | 0.0099 | | | | | 0.308 | 0.0383 | 141 |
| NTAP 4 | 0.9144*** | 0.4483 | $-5.705\times10^{-10}$ | 0.0060 | | | | | 0.381 | 0.0152 | 247 |
| WDC | 1.7123*** | 1.3365** | $2.571\times10^{-10}$ | −0.0224** | 0.0145** | 0.0079 | 0.0184*** | −0.0633*** | 0.330 | | 518 |
| WDC | 1.6933*** | 1.1960** | $5.886\times10^{-11}$ | −0.0029 | | | | | 0.314 | 0.0243 | 518 |
| WDC 1 | 2.0985*** | 1.8179 | $2.527\times10^{-10}$ | −0.0039 | | | | | 0.405 | 0.0239 | 68 |
| WDC 2 | 0.9871*** | 0.6789 | $3.199\times10^{-10}$ | −0.0117 | | | | | 0.137 | 0.0181 | 165 |
| WDC 3 | 1.5196*** | 0.2117 | $2.541\times10^{-11}$ | 0.0023 | | | | | 0.281 | 0.0279 | 279 |
| WDC 4 | −0.6928 | 48.0119 | $8.676\times10^{-9}$ | −0.8526 | | | | | 0.201 | 0.0294 | 6 |

*(Continued)*

Table 5. (Continued)

| | S&P 500 return | return momentum | volume | constant | state 1 | state 2 | state 3 | state 4 | Adjusted R squared | $\gamma$ | N for each state |
|---|---|---|---|---|---|---|---|---|---|---|---|
| **Solar Weight: 0.21%** | | | | | | | | | | | |
| FSLR | 1.2906*** | −1.3966 | $8.216 \times 10^{-10}$ | −0.0071 | −0.0014 | 0.0044 | −0.0072 | −0.0030 | 0.183 | | 518 |
| FSLR | 1.2948*** | −1.3741 | $7.679 \times 10^{-10}$* | −0.0079 | | | | | 0.185 | 0.0152 | 518 |
| FSLR 1 | 1.4889*** | −1.2747 | $-2.217 \times 10^{-9}$** | 0.0139* | | | | | 0.398 | 0.0177 | 201 |
| FSLR 2 | 1.3217** | 2.3345 | $1.527 \times 10^{-9}$ | −0.0079 | | | | | 0.090 | 0.0242 | 87 |
| FSLR 3 | −4.2187 | −16.3884 | $3.891 \times 10^{-10}$ | −0.0048 | | | | | 0.036 | 0.0113 | 25 |
| FSLR 4 | 1.2570*** | 0.6608 | $2.437 \times 10^{-9}$*** | −0.0258** | | | | | 0.260 | 0.0093 | 205 |
| ENPH | 2.2239*** | 0.0171 | $2.205 \times 10^{-9}$*** | −0.0332*** | −0.0521** | 0.0276** | −0.0005 | −0.0081 | 0.174 | | 518 |
| ENPH | 2.1323*** | 0.1182 | $8.517 \times 10^{-10}$** | −0.0067 | | | | | 0.166 | 0.1949 | 518 |
| ENPH 1 | 5.0399** | −1.5779 | $6.105 \times 10^{-9}$ | −0.2522* | | | | | 0.169 | 0.0393 | 25 |
| ENPH 2 | 2.7576*** | 0.4809 | $6.615 \times 10^{-9}$*** | −0.0282** | | | | | 0.165 | 0.4517 | 206 |
| ENPH 3 | 1.7899*** | −1.4583 | $2.138 \times 10^{-9}$** | −0.0337** | | | | | 0.276 | 0.0070 | 217 |
| ENPH 4 | 1.9763*** | 0.3376 | $6.84 \times 10^{-10}$ | 0.0009 | | | | | 0.141 | 0.0615 | 70 |
| **Electronics & Computer Distribution Weight: 0.16%** | | | | | | | | | | | |
| SNX | 1.4279*** | 1.8082 | $-2.042 \times 10^{-9}$ | 0.0138** | −0.0105*** | −0.0107*** | 0.0192 | −0.0158*** | 0.426 | | 518 |
| SNX | 1.3700*** | 0.4659 | $1.075 \times 10^{-9}$ | −0.0022 | | | | | 0.410 | 0.0169 | 518 |
| SNX 1 | 1.4106*** | −0.7270 | $-3.68 \times 10^{-9}$** | 0.0066 | | | | | 0.355 | 0.0217 | 247 |
| SNX 2 | 1.1628*** | −1.5860 | $-3.978 \times 10^{-9}$* | 0.0062 | | | | | 0.490 | 0.0101 | 206 |
| SNX 3 | 1.2489 | −69.3548 | $8.727 \times 10^{-9}$ | −0.0935 | | | | | −0.045 | 0.0209 | 11 |
| SNX 4 | 1.7735*** | 4.2523* | $-6.266 \times 10^{-9}$ | 0.0559** | | | | | 0.557 | 0.0191 | 54 |
| ARW | 1.2503*** | 1.5397** | $3.11 \times 10^{-9}$ | −0.0154 | 0.0068 | 0.0081 | 0.0085 | −0.0388 | 0.436 | | 518 |
| ARW | 1.2442*** | 1.4486** | $2.21 \times 10^{-9}$ | −0.0057 | | | | | 0.437 | 0.0120 | 518 |
| ARW 1 | 1.3123*** | −6.3683** | $1.13 \times 10^{-9}$ | −0.0053 | | | | | 0.501 | 0.0144 | 248 |
| ARW 2 | 0.9345*** | −0.6362 | $-4.247 \times 10^{-10}$ | 0.0014 | | | | | 0.337 | 0.0090 | 240 |
| ARW 3 | 1.8488*** | 1.3481 | $2.948 \times 10^{-8}$* | −0.1100 | | | | | 0.578 | 0.0164 | 29 |
| ARW 4 | Nan | Nan | Nan | Nan | | | | | Nan | 0.0085 | 1 |

Note: *, ** and *** indicate the significance level at 10%, 5% and 1% respectively.

behaviors. Our findings depict investor sentiment to general market regime shift dominates the volume effect under most cases. We show investor sentiment differs significantly across different stock market regimes for most tech stocks with large and medium market shares. While volume effect also plays important role in determinations of tech companies stock returns, the salient investor sentiment effect even dominants the volume effect among large tech companies and individual sectors. By controlling the investment sentiment effect, the volume effect can be fully revealed. Despite of some edge states which exposed to little investor sentiment or volume effect, most of the stock returns depicts investment effect and volume effect at whole stock level and under individual states. In contrast, the momentum effect is comparatively trivial on the whole stock level and individual state level. Above empirical findings coherent with tech sector's fast growth rate, high macroeconomic factors driven and strong policy dependence properties.

In the future research, we would extend our study into multiple dimensions. We would like to list following three aspects as some potential topics we would like to cover in our future research. (1) In our study, we have classified the stock market time series into individual states and analyze the reaction of the stock return to the investor sentiment. The model has helped us filter out the edge cases in the time series, especially for those states which contain very few observations. We would like to look deeply of these states and analyze the specific uniqueness of these states to fully understand the PCA-HMM technique. (2) Although we successfully perform sector analysis and identify the impact of investor sentiment on stock returns, we can further investigate the relationships between investor sentiment among these states with original principal component factor loadings to identify the inner relationships between different financial or macroeconomic factors and investor sentiment. (3) In our next step studies, we can include the IPO effect into our model to discuss the IPO effect on stock returns. As we seen, IPO is a positive signal to the market which can gather investment and attract market attention. The IPO effect can well represent market perceptions of the company's future growth potential. We will adopt our PCA-HMM model to study IPO data and calibrate investor sentiment change due to IPO.

## Author contributions

**Data curation:** Shaoshu Li.

**Formal analysis:** Shaoshu Li.

**Methodology:** Shaoshu Li.

**Validation:** Shaoshu Li.

**Visualization:** Shaoshu Li.

**Writing – original draft:** Shaoshu Li.

**Writing – review & editing:** Shaoshu Li.

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
