## [Decision Letter · Decision Letter 0]

21 May 2025

Dear Dr. Li,

Thank you for submitting your manuscript to PLOS ONE. After careful consideration, we feel that it has merit but does not fully meet PLOS ONE’s publication criteria as it currently stands. Therefore, we invite you to submit a revised version of the manuscript that addresses the points raised during the review process.

We look forward to receiving your revised manuscript.

Kind regards,

Academic Editor

PLOS ONE

Journal Requirements:

Reviewers' comments:

Reviewer's Responses to Questions

**Comments to the Author**

1. Is the manuscript technically sound, and do the data support the conclusions?

Reviewer #1: Yes

2. Has the statistical analysis been performed appropriately and rigorously?

Reviewer #1: Yes

3. Have the authors made all data underlying the findings in their manuscript fully available?

Reviewer #1: Yes

4. Is the manuscript presented in an intelligible fashion and written in standard English?

Reviewer #1: Yes

Reviewer #1: Both citations and references should be reviewed to include recent literature from 2020 - 2025.

The manuscript should be updated and upgraded. Discussion of findings should be link to what is applicable in reality, with practical implications.

**Do you want your identity to be public for this peer review?** For information about this choice, including consent withdrawal, please see our Privacy Policy

Reviewer #1: **Yes: ** Stephen Alaba John, PhD

---

## [Author Response · Author response to Decision Letter 1]

26 Jun 2025

Thank you so much reviewers for your kind comments so far! Very grateful!

---

## [Editor Report · Decision Letter 1]

22 Jul 2025

Dear Dr. Li,

Thank you for submitting your manuscript to PLOS ONE. After careful consideration, we feel that it has merit but does not fully meet PLOS ONE’s publication criteria as it currently stands. Therefore, we invite you to submit a revised version of the manuscript that addresses the points raised during the review process.

We look forward to receiving your revised manuscript.

Kind regards,

Yao Zheng

Academic Editor

PLOS ONE

Journal Requirements:

Additional Editor Comments:

In your response to the reviewer: can you provide a detailed reply to the following questions:

Both citations and references should be reviewed to include recent literature from 2020-2025.

The manuscript should be updated and improved. The discussion of findings should connect to real-world applications and include practical implications.

Please list all the updates you have made in your revision.

Regards,

Associate Editor

---

## [Author Response · Author response to Decision Letter 2]

27 Jul 2025

Dear Reviewer, we have improved our work according to your kind suggestions and the detail improvements are shown in the response letter attached. Thank you so much for your great help!

---

## [Editor Report · Decision Letter 2]

20 Aug 2025

Momentum, Volume and Investor Sentiment Study for U.S. Technology Sector Stocks ——A Hidden Markov Model based Principal Component Analysis

PONE-D-25-06876R2

Dear Dr. Li,

We’re pleased to inform you that your manuscript has been judged scientifically suitable for publication and will be formally accepted for publication once it meets all outstanding technical requirements.

Kind regards,

Academic Editor

PLOS ONE
---

## [Editor Report · Acceptance letter]

PONE-D-25-06876R2

PLOS ONE

Dear Dr. Li,

I'm pleased to inform you that your manuscript has been deemed suitable for publication in PLOS ONE. Congratulations! Your manuscript is now being handed over to our production team.

Kind regards,

on behalf of

Dr. Yao Zheng

Academic Editor

PLOS ONE